# Product Ranking for Revenue Maximization with Multiple Purchases

**Renzhe Xu**[1], **Xingxuan Zhang**[1], **Bo Li**[2], **Yafeng Zhang**[3], **Xiaolong Chen**[3], **Peng Cui**[1*]
[1]Department of Computer Science and Technology, Tsinghua University, Beijing, China
[2]School of Economics and Management, Tsinghua University, Beijing, China
[3]Meituan Inc., Beijing, China
xrz199721gmail.com, xingxuanzhang@hotmail.com, libo@sem.tsinghua.edu.cn
yafengz@outlook.com, chenxiaolong13@meituan.com, cuip@tsinghua.edu.cn

## Abstract

Product ranking is the core problem for revenue-maximizing online retailers. To design proper product ranking algorithms, various consumer choice models are proposed to characterize the consumers' behaviors when they are provided with a list of products. However, existing works assume that each consumer purchases at most one product or will keep viewing the product list after purchasing a product, which does not agree with the common practice in real scenarios. In this paper, we assume that each consumer can purchase multiple products at will. To model consumers' willingness to view and purchase, we set a random attention span and purchase budget, which determines the maximal amount of products that he/she views and purchases, respectively. Under this setting, we first design an optimal ranking policy when the online retailer can precisely model consumers' behaviors. Based on the policy, we further develop the Multiple-Purchase-with-Budget UCB (MPB-UCB) algorithms with $\tilde{O}(\sqrt{T})$ regret that estimate consumers' behaviors and maximize revenue simultaneously in online settings. Experiments on both synthetic and semi-synthetic datasets prove the effectiveness of the proposed algorithms. The source code is available at https://github.com/windxrz/MPB-UCB.

## 1 Introduction

Online retailing has become increasingly popular over the last decades [17, 28, 52]. The way of product ranking is the crux for online retailers because it determines the consumers' shopping behaviors [17] and thus influences the retailers' revenue [20, 49]. For instance, the probability of consumers' purchasing from a firm or clicking an advertisement is strongly related to the display order [8, 3, 33]. Therefore, it is crucial for online retailers to design proper product ranking algorithms for revenue maximization.

The consumer choice model, when faced with a list of products, is the basis of the problem [17]. Generally speaking, existing works assume that the consumers view the list of products sequentially and select products according to a choice model [43, 42, 32, 38, 18, 26, 25, 37]. However, most of them assume that each consumer purchases at most one product and stop browsing immediately afterward, which does not agree with the common practice since a consumer may expect multiple purchases on a website or an app [34, 35, 16, 28]. Recently, several works consider a multiple-purchase setting while their targets are not revenue maximization [28, 15], or they assume that consumers always continue viewing the product list after purchasing any product, which is not practical in real scenarios [40].

---

[*]Corresponding Author

In this paper, we propose a more realistic consumer choice model to characterize consumer behaviors under multiple-purchase settings. Following [13, 17, 40], we assume that the consumers view the product list sequentially and purchase a product when its utility exceeds a certain threshold. Similar to [17], we assume the existence of the attention span for each consumer, which determines the maximal amount of products that the consumer is willing to view. More importantly, instead of forcing the consumer to quit viewing once she purchases one product, we allow each consumer to conduct multiple purchases at will and explore product ranking models for revenue maximization. Specifically, we constrain each consumer with a purchase budget to determine the maximal amount of purchases. Both the attention span and the purchase budget are assumed to obey the geometric distribution [13, 40].

Under this setting, we study the optimal product ranking policy to maximize online retailers' revenue. We first characterize the optimal ranking policy that achieves the maximal revenue when the online retailer knows the consumer parameters, including the purchase probabilities for each product and the distribution of the attention span and purchase budget. Afterward, we investigate the problem in online learning frameworks where consumer parameters are unknown to online retailers. In both non-contextual (*i.e.*, all consumers share the same parameters) and contextual settings (*i.e.*, consumers have personalized parameters), we provide the Multiple-Purchase-with-Budget UCB (MPB-UCB) algorithms that could fit consumers' parameters and maximize revenue simultaneously. We further theoretically prove that the algorithms achieve $\tilde{O}(\sqrt{T})$ regrets in both settings. Finally, we conduct experiments to verify the effectiveness of the algorithms.

To conclude, our contributions are highlighted as follows:

- We propose a novel consumer choice model to deal with multiple-purchase activities of consumers in online retailing scenarios.
- We characterize the optimal ranking policy and further design the MPB-UCB algorithms with $\tilde{O}(\sqrt{T})$ regret based on them in both non-contextual and contextual online settings.
- We conduct experiments on both synthetic and semi-synthetic datasets to prove the effectiveness of our method in the multiple-purchase setting.

## 2 Related Works

**Consumer choice model**   Traditional choice models adopt the multinomial logic (MNL) [12, 22, 43] models to characterize consumers' behaviors. However, these models can not describe the behaviors when the consumers are provided with a list of products. As a result, several choice models are proposed recently. In detail, Abeliuk et al. [2] add a position bias term in the traditional MNL model. Gallego et al. [30], Aouad and Segev [4] suppose that the products are framed into a set of virtual web pages. Consumers select a product, if any, from these pages following a general choice model (such as MNL). Flores et al. [29] propose the Sequential Multinomial Logit (SML), which assumed a two-stage MNL model and supposed that products are divided into two stages with prior knowledge. Liu et al. [41], Feldman and Segev [27], Gao et al. [31] further extend the setting to multiple stages. These works suppose that each stage contains multiple products while recently several works assume that a product is treated as a single stage [17, 13, 14]. These models are also closely related to the cascade model [12, 36, 38, 54, 19]. Gao et al. [32] further propose the general cascade click models. Following these works, we suppose that each stage contains only one product and the consumers view the products sequentially.

In addition, several works consider the setting where consumers may have multiple interactions with the platforms. However, most of them focus on the click model [15, 16, 34–36] or MNL-based consumer choice models [48, 53, 50]. In addition, Ferreira et al. [28] maximize the number of consumers who engage with the site. Liang et al. [40] aim to maximize the revenue in the multiple-purchase setting while they assume that consumers always continue viewing the product list after purchasing any product, which is not practical in real scenarios.

**Online learning and multi-armed bandit**   Online retailers need to design proper algorithms to model consumers' behaviors and maximize revenue at the same time, which is closely related to the multi-armed bandit framework [47, 45]. Many works [5, 9, 11, 10, 24] have been proposed to deal with demand learning or price experimentation problems with the help of the framework in the field

of revenue management. Recently, more methods [40, 17, 13, 14, 6, 44] adopt similar techniques in the product ranking setting, which inspires the algorithm proposed in this paper.

## 3 Preliminaries

Let $[N] \triangleq \{1, 2, \ldots, N\}$ be the set of all products owned by an online retailer (he). The product $k$ generates revenue $r_k$ when a consumer purchases it. Let $r_{\max} \triangleq \max_{k \in [N]} r_k$ be the maximal revenue for all products and $\boldsymbol{r} = \{r_k\}_{k=1}^N$. We use the notation $\boldsymbol{\sigma}_t \dashv [N]$ to denote the ranking policy $\boldsymbol{\sigma}_t$ on the products for the $t$-th consumer (she). More precisely, $\forall k \in [N]$, $\sigma_{t,k} \in [N]$ represents the product displayed in the $k$-th position for her. Reversely, $\sigma_{t,k}^{-1}$ denotes the position of the $k$-th product in the ranking list. Furthermore, for any subset $Q \subseteq [N]$ that represents the possible positions in the ranking list, we use $\sigma_{t,Q}$ to denote the corresponding set of the products $w.r.t.$ the position set $Q$, $i.e.$, $\sigma_{t,Q} = \{\sigma_{t,k} : k \in Q\}$. In addition, we use $\max(\cdot)$ to denote the maximal element in a set.

**Consumer choice model for multiple purchases**  When the $t$-th consumer arrives, she views the list of products sequentially. We assume that the behaviors of different consumers are independent. The probability of product $k$ being purchased by her is denoted as $\lambda_{t,k}$ and let $\boldsymbol{\lambda}_t = \{\lambda_{t,k}\}_{k=1}^N$. We assume for tractability that consumer interests in any products are independent, which is common in product ranking papers [4, 18, 23, 49]. This indicates that the purchase behavior of any product set does not affect the purchase probabilities of other products.

To characterize the consumers' behaviors under multiple-purchase settings, we assume that the consumer is endowed with the attention span $V_t$ and purchase budget $B_t$, which determine the maximal number of the products that she will view and purchase, respectively. In reality, the consumer always has random attention span and purchase budget [17]. Therefore, following [13, 40], we assume $V_t \sim \mathrm{Geo}(1 - q_t)$ and $B_t \sim \mathrm{Geo}(1 - s_t)$ are independent geometric distributions with parameters $1 - q_t$ and $1 - s_t$, respectively. As a result, $P(V_t = v) = q_t^{v-1}(1 - q_t)$ for any $v \geq 1$ and $P(B_t = b) = s_t^{b-1}(1 - s_t)$ for any $b \geq 1$. With attention span $V_t$ and purchase budget $B_t$, the consumer keeps viewing the products until she views $V_t$ or purchases $B_t$ products.

Because of the attention span $V_t$, the consumer will not purchase products whose positions are greater than $V_t$ in the ranking list. Let $Q \subseteq [V_t]$ denote the possible positions of the purchased products by the consumer. The corresponding indices of purchased products are $\sigma_{t,Q}$. Therefore, the probability that the consumer purchases the product set $\sigma_{t,Q}$ is given by

$$P^{\mathrm{buy}}(Q; V_t, B_t, \boldsymbol{\lambda}_t, \boldsymbol{\sigma}_t) = \begin{cases} \left(\prod_{k \in Q} \lambda_{t,\sigma_{t,k}}\right)\left(\prod_{k \in [V_t] \setminus Q} \left(1 - \lambda_{t,\sigma_{t,k}}\right)\right), & \text{if } |Q| < B_t. \\ \left(\prod_{k \in Q} \lambda_{t,\sigma_{t,k}}\right)\left(\prod_{k \in [V_t] \setminus Q, k \leq \max(Q)} \left(1 - \lambda_{t,\sigma_{t,k}}\right)\right), & \text{if } |Q| = B_t. \\ 0, & \text{if } |Q| > B_t. \end{cases} \quad (1)$$

Specifically, the probability $P^{\mathrm{buy}}(Q; V_t, B_t, \boldsymbol{\lambda}_t, \boldsymbol{\sigma}_t)$ is calculated based on the cardinality of $Q$. When $|Q| < B_t$, the consumer does not meet her purchase budget and stops browsing the list after viewing $V_t$ products. When $|Q| = B_t$, the consumer stops browsing as long as she purchases the last product in the list $Q$. When $|Q| > B_t$, the probability becomes 0 since the number of purchases $|Q|$ exceeds her purchase budget.

**Remark.**  The setting of the purchase budget provides a rational way to characterize the consumers' behaviors with multiple purchases and extends previous works that suppose consumers would purchase at most one product [13, 17]. Furthermore, when $s_t \to 0$, the consumers would tend to purchase at most one product and our setting degenerates to the setting proposed in [13]. Thus the previous setting can be considered a special case of our setting. In addition, Liang et al. [40] assume that consumers always continue viewing the product list after purchasing any product, which is not practical in real scenarios. By contrast, the purchase budget proposed in our paper leads to a different and more practical consumer choice model. In addition, our model introduces an extra parameter ($i.e.$, the parameter), which makes our problem more challenging.

**Revenue optimization**  The online retailer chooses a ranking policy $\boldsymbol{\sigma}_t \dashv [N]$ on the products for the $t$-th consumer. For the consumer with fixed purchase probabilities for each product $\boldsymbol{\lambda}_t$, attention

span $V_t$, and purchase budget $B_t$, the expected revenue could be written as

$$R(V_t, B_t; \boldsymbol{\lambda}_t, \boldsymbol{\sigma}_t) = \sum_{Q \subseteq [V_t]} P^{\text{buy}}(Q; V_t, B_t, \boldsymbol{\lambda}_t, \boldsymbol{\sigma}_t) \left( \sum_{i \in Q} r_{\sigma_{t,i}} \right). \tag{2}$$

In reality, the attention span $V_t$ and purchase budget $B_t$ are random and can not be estimated accurately by the online retailer. As a result, the retailer calculates the expected revenue for the $t$-th consumer by sampling $V_t$ from $\text{Geo}(1 - q_t)$ and sampling $B_t$ from $\text{Geo}(1 - s_t)$, *i.e.*,

$$\text{Revenue}\,(\boldsymbol{\sigma}_t; \boldsymbol{\lambda}_t, q_t, s_t) = \mathbb{E}_{V_t \sim \text{Geo}(1-q_t), B_t \sim \text{Geo}(1-s_t)} \left[ R(V_t, B_t; \boldsymbol{\lambda}_t, \boldsymbol{\sigma}_t) \right]. \tag{3}$$

To maximize the revenue, the optimal ranking policy is given by

$$\boldsymbol{\sigma}_t^*(\boldsymbol{\lambda}_t, q_t, s_t) = \underset{\boldsymbol{\sigma}_t \dashv [N]}{\arg\max}\, \text{Revenue}\,(\boldsymbol{\sigma}_t; \boldsymbol{\lambda}_t, q_t, s_t). \tag{4}$$

For simplicity, we use $\boldsymbol{\sigma}_t^*$ to denote $\boldsymbol{\sigma}_t^*(\boldsymbol{\lambda}_t, q_t, s_t)$.

**Basic assumption** We also need the following assumption to guarantee the tractability of the proposed problem. In reality, consumers always have finite attention span, which indicates that $q_t$ is always smaller than 1.

**Assumption 3.1.** There exists a known constant $\epsilon_Q > 0$ such that for all $t > 0$, $q_t \leq 1 - \epsilon_Q$.

## 4 Optimal Ranking Policy Given Consumers' Characteristics

In this section, we introduce the optimal ranking policy when the online retailer exactly knows the consumers' parameters, including the purchase probability for each product $\boldsymbol{\lambda}$, random attention span (parametrized with $q$), and random purchase budget (parametrized with $s$).

We drop the subscript $t$ in this section for simplicity. The optimal ranking strategy stems from the following theorem.

**Theorem 4.1.** *Under Assumption 3.1, the following holds for the optimal ranking strategy* $\boldsymbol{\sigma}^*(\boldsymbol{\lambda}, q, s)$.

$$\forall 1 \leq i < N, \quad \frac{\lambda_{\sigma_i^*} r_{\sigma_i^*}}{1 - q + q(1-s)\lambda_{\sigma_i^*}} \geq \frac{\lambda_{\sigma_{i+1}^*} r_{\sigma_{i+1}^*}}{1 - q + q(1-s)\lambda_{\sigma_{i+1}^*}}. \tag{5}$$

**Remark.** The expected revenue of a single product $\sigma_i^*$ is $\lambda_{\sigma_i^*} r_{\sigma_i^*}$, which grows as the purchase probability and the revenue generated by the product increase. Yet purchasing a product reduces the purchase probability of the following products, and thus the expected revenue of other products decreases as $\lambda_{\sigma_i^*}$ increases. Furthermore, when $s \to 0$, the term in Equation (5) becomes the same as that in [13, Theorem 1] if ignoring the marketing fatigue (*i.e.*, overexposure to unwanted marketing messages) term in their result. As a result, Theorem 4.1 consistently extends the results on the single-purchase setting [13] to the multiple-purchase setting.

In addition, Liang et al. [40] assume that the online retailer will stop recommending products with a certain probability $1 - w_t$ to avoid the cost brought by marketing fatigue. As a result, the parameter $w_t$ is a part of the ranking policy instead of consumers' characteristics and should be optimized to achieve maximal revenue. In our paper, we focus on the multiple purchases with budget setting and leave the extension to the case with the marketing fatigue as future work.

According to Theorem 4.1, the online retailer can provide the optimal ranking policy $\boldsymbol{\sigma}^*$ by sorting the products in descending order with value $(\lambda_k r_k) / (1 - q + q(1-s)\lambda_k)$ for the $k$-th product. The time complexity is $O(N \log N)$.

## 5 Online Learning of the Ranking Policy

In this section, we consider a more realistic scenario where the online retailer has no prior knowledge about consumers' characteristics. At each timestamp, a consumer arrives and the online retailer can only provide proper product ranking policies based on the feedback of previous consumers. As a result, the retailer needs to design online learning algorithms to model consumers' behaviors and maximize revenue in the meantime. We consider two settings (*i.e.*, the non-contextual setting where all consumers share the same parameters (Section 5.1) and the contextual setting where consumers have personalized behaviors (Section 5.2)) and develop the Multiple-Purchase-with-Budget UCB (MPB-UCB) algorithms on both settings. Before introducing the details of the algorithms, we first formally define the notations to characterize the consumers' behaviors.

**Notations for consumers' behaviors**   Let $\Phi_t \in \mathbb{Z}_+$ be the random variable that denotes the number of products viewed by the $t$-th consumer. Let random variable $\gamma_{t,k} \in \{0,1\}$ denote whether the consumer purchases the product displayed in the $k$-th position (*i.e.*, product $\sigma_{t,k}$) if she views the product. Specifically, $\gamma_{t,k} = 1$ if and only if the product is purchased. Let $\boldsymbol{\gamma}_t = \{\gamma_{t,k}\}_{k=1}^{\Phi_t}$.

Let random variable $\eta_{t,k} \in \{0,1\}$ denote whether the consumer continues viewing products if she views the product displayed in the $k$-th position but does not buy it. $\eta_{t,k}$ is observed only when $k \leq \Phi_t$ and $\gamma_{t,k} = 0$. Let random variable $\mu_{t,k} \in \{0,1\}$ denote whether the consumer keeps viewing products if she buys the product in the $k$-th position. $\mu_{t,k}$ is observed only when $k \leq \Phi_t$ and $\gamma_{t,k} = 1$. Let $\boldsymbol{\eta}_t = \{\eta_{t,k}\}_{k=1}^{\Phi_t}$ and $\boldsymbol{\mu}_t = \{\mu_{t,k}\}_{k=1}^{\Phi_t}$.

Let $\mathrm{Ber}(\cdot)$ denote the Bernoulli distribution and we have the following result.

**Lemma 5.1.** *$\{\gamma_{t,k}\}_{t>0,1\leq k<N}$ are independent of each other, $\{\eta_{t,k}\}_{t>0,1\leq k<N}$ are independent of each other, and $\{\mu_{t,k}\}_{t>0,1\leq k<N}$ are independent of each other. In addition, $\forall t > 0$ and $1 \leq k < N$, $\gamma_{t,k} \sim \mathrm{Ber}\left(\lambda_{\sigma_{t,k}}\right)$, $\eta_{t,k} \sim \mathrm{Ber}(q_t)$, and $\mu_{t,k} \sim \mathrm{Ber}(q_t s_t)$.*

Lemma 5.1 is the basis of the algorithms that estimate parameters $\boldsymbol{\lambda}$, $q$, and $s$ from historical data. In addition, compared with $s_t$, it is easier to estimate $q_t s_t$ from data and we denote it as $w_t \triangleq q_t s_t$.

## 5.1   Non-contextual Online Setting

In this subsection, we assume that all consumers share the same parameters, including the purchase probabilities on different products and the parameters on the distribution of the attention span and purchase budget. In detail, we assume that there exist $\boldsymbol{\lambda}, q, s$ such that $\boldsymbol{\lambda}_t = \boldsymbol{\lambda}$, $q_t = q$, and $s_t = s$.

### A)   Estimation of Parameters

To estimate the parameters at timestamp $t$, we leverage the observed behavior of consumers before timestamp $t$. We first calculate the following statistics that can help provide unbiased estimators of the consumer parameters.

In detail, let $C_{t,k}$ be the number of times that the product $k$ is observed and $c_{t,k}$ be the number of times the product $k$ is purchased, *i.e.*, $C_{t,k} \triangleq \sum_{u=1}^{t}\sum_{i=1}^{\Phi_t} \mathbb{I}\left[\sigma_{t,i} = k\right]$ and $c_{t,k} \triangleq \sum_{u=1}^{t}\sum_{i=1}^{\Phi_t} \mathbb{I}\left[\sigma_{t,i} = k\right]\mathbb{I}\left[\gamma_{t,i} = 1\right]$. In addition, Let $D_t^Q$ be the number of times that consumers view a product and do not purchase it and $d_t^Q$ be the number of times that consumers continue viewing the list after viewing a product without purchasing it, *i.e.*, $D_t^Q \triangleq \sum_{u=1}^{t}\sum_{i=1}^{\Phi_t} \mathbb{I}\left[\gamma_{t,i} = 0\right]$ and $d_t^Q \triangleq \sum_{u=1}^{t}\sum_{i=1}^{\Phi_t} \mathbb{I}\left[\gamma_{t,i} = 0\right]\eta_{t,i}$. Furthermore, let $D_t^W$ be the number of times that consumers purchase a product and $d_t^W$ be the number of times that consumers continue viewing the list after purchasing a product, *i.e.*, $D_t^W \triangleq \sum_{u=1}^{t}\sum_{i=1}^{\Phi_t} \mathbb{I}\left[\gamma_{t,i} = 1\right]$ and $d_t^W \triangleq \sum_{u=1}^{t}\sum_{i=1}^{\Phi_t} \mathbb{I}\left[\gamma_{t,i} = 1\right]\mu_{t,i}$. With these statistics, the parameters $\boldsymbol{\lambda}$, $q$, and $w = qs$ are estimated as follows.

$$\hat{\lambda}_{t,k} \triangleq c_{t,k}/C_{t,k}, \quad \hat{q}_t \triangleq d_t^Q/D_t^Q, \quad \hat{w}_t \triangleq d_t^W/D_t^W. \tag{6}$$

We provide the following proposition that gives the error bound of our parameter estimation approach.

**Proposition 5.2.** *For all $t \in [T]$, with probability at least $1 - 6Nt^{-3}$, the following holds:*

$$\forall k \in [N], \left|\hat{\lambda}_{t,k} - \lambda_k\right| \leq \sqrt{\frac{2\log t}{C_{t,k}}}, \quad |\hat{q}_t - q| \leq \sqrt{\frac{2\log t}{D_t^Q}}, \quad \text{and} \quad |\hat{w}_t - qs| \leq \sqrt{\frac{2\log t}{D_t^W}}. \tag{7}$$

### B)   Algorithm

Following the classic *optimism in the face of uncertainty principle* [1], we design a UCB-like algorithm that learns consumer behaviors and maximizes revenue simultaneously. In detail, according to Proposition 5.2, we learn the optimistic estimators of the parameters given as follows

$$\tilde{\lambda}_{t,k} \triangleq \min\left\{1, \hat{\lambda}_{t,k} + \sqrt{\frac{2\log t}{C_{t,k}}}\right\}, \tilde{q}_t \triangleq \min\left\{1 - \epsilon_Q, \hat{q}_t + \sqrt{\frac{2\log t}{D_t^Q}}\right\}, \tilde{w}_t \triangleq \min\left\{\tilde{q}_t, \hat{w}_t + \sqrt{\frac{2\log t}{D_t^W}}\right\}. \tag{8}$$

**Algorithm 1:** MPB-UCB (Non-contextual)

---

1 **Input:** Products revenue $r$ and hyper-parameter $\epsilon_Q$.
2 **Initialization:** $\tilde{\lambda}_{0,k} = 1$ for $k \in [N]$, $\tilde{q}_0 = 1 - \epsilon_Q$, $\tilde{w}_0 = 1 - \epsilon_Q$.
3 **for** *t=1:T* **do**
4      Let $\boldsymbol{\sigma}_t$ be the optimal offline policy from Theorem 4.1 with $\boldsymbol{\lambda} = \tilde{\boldsymbol{\lambda}}_{t-1}$, $q = \tilde{q}_{t-1}$, and
        $s = \tilde{w}_{t-1}/\tilde{q}_{t-1}$.
5      Offer ranking policy $\boldsymbol{\sigma}_t$ and observe $\Phi_t, \boldsymbol{\gamma}_t, \boldsymbol{\eta}_t, \boldsymbol{\mu}_t$.
6      Update statistics $C_{t,k}, c_{t,k}, D_t^Q, d_t^Q, D_t^W$, and $d_t^W$.
7      Calculate $\tilde{\lambda}_{t,k}, \tilde{q}_t$, and $\tilde{w}_t$ by Equations (6) and (8).
8 **end**

---

As shown in Algorithm 1, when the $t$-th consumer arrives, the online retailer displays the products based on Theorem 4.1 with $\tilde{\boldsymbol{\lambda}}_{t-1}, \tilde{q}_{t-1}$, and $\tilde{w}_{t-1}/\tilde{q}_{t-1}$ shown in Line 4. Afterward, the online retailer updates the statistics according to consumers' feedback as shown in Lines 5 and 6. The estimators are then updated in Line 7.

## C) Regret Analysis

To theoretically analyze the performance of Algorithm 1, we first introduce the regret, which measures the total difference between the maximal revenue achieved by $\boldsymbol{\sigma}^*$ and the cumulative reward by an online algorithm after $T$ rounds.

$$\text{Reg}_T \triangleq \sum_{t=1}^{T} \text{Revenue}\left(\boldsymbol{\sigma}^*; \boldsymbol{\lambda}, q, s\right) - \text{Revenue}\left(\boldsymbol{\sigma}_t; \boldsymbol{\lambda}, q, s\right). \tag{9}$$

$\text{Reg}_T$ is a random variable and the randomness comes from the uncertainty in consumers' behaviors. As a result, we focus on $\mathbb{E}\left[\text{Reg}_T\right]$, which is a common routine in bandit literature [46]. The performance of the algorithm is given by the following theorem.

**Theorem 5.3.** *Under Assumption 3.1 (with parameter $\epsilon_Q$), for any $T > 1$, the regret achieved by Algorithm 1 is bounded by*

$$\mathbb{E}\left[\text{Reg}_T\right] \leq C \cdot \frac{r_{\max}}{\epsilon_Q} N \sqrt{T \log T}, \tag{10}$$

*where $C$ is a absolute constant (independent of problem parameters) and $r_{\max} = \max_{k \in [N]} r_k$.*

**Remark.** We analyze different parameters' impacts on the regret bound in Equation (10). Firstly, the regret grows at $\tilde{O}(\sqrt{T})$, which is a standard result in online learning algorithms [46]. Secondly, the regret bound is linearly related to $N$, which is similar to the results in cascade bandit models [54] and revenue management literature [17]. Finally, the bound depends on $r_{\max}/\epsilon_Q$, which measures the maximal expected revenue the online retailer could achieve from each consumer.

## 5.2 Contextual Online Setting

We consider a more realistic scenario where the consumers' behaviors are various and depend on their own features. In detail, let $x_t \in \mathbb{R}^{m_x}$ be the feature of the $t$-th consumer with dimension $m_x$ and $y_{t,k} \in \mathbb{R}^{m_y}$ be the joint feature of the $t$-th consumer and $k$-th product with dimension $m_y$. $y_{t,k}$ can be the concatenation of consumer features and product features or fusion through a transformation such as a matrix multiplication. In this paper, we adopt concatenation for simplicity. Let $\|\cdot\|$ denote the $\ell_2$ norm. Following [17], we consider the classic linear bandit setting [7, 21, 39] and the ground-truth data-generating process is given by the following assumptions.

**Assumption 5.1.** There exist constant vectors $\beta^\Lambda \in \mathbb{R}^{m_y}$, $\beta^Q \in \mathbb{R}^{m_x}$, and $\beta^S \in \mathbb{R}^{m_x}$, such that

$$\lambda_{t,k} = y_{t,k}^\top \beta^\Lambda, \forall k \in [N], \quad q_t = x_t^\top \beta^Q, \quad \text{and} \quad s_t = x_t^\top \beta^S. \tag{11}$$

**Assumption 5.2.** For any $t > 0$ and $k \in [N]$, $\|x_t\| \leq 1$ and $\|y_{t,k}\| \leq 1$. In addition, there exists a constant $U > 0$ such that $\|\beta^\Lambda\|, \|\beta^Q\|, \|\beta^S\| \leq U$.

**Remark.** These two assumptions are common in revenue maximization and linear bandits [1, 46]. Assumption 5.1 can be satisfied with kernel functions for complex data-generating processes and Assumption 5.2 can be achieved by normalization. Thus the assumptions are rational and easily attainable in real applications.

## A)  Estimation of Parameters

For the estimation of $\beta^\Lambda$ and $\beta^Q$, we consider the ridge regression method. Given the observation of consumer's behaviors till the $t$-th timestamp, the estimation $\hat{\beta}_t^\Lambda$, $\hat{\beta}_t^Q$ is given by

$$\hat{\beta}_t^\Lambda = \underset{\beta \in \mathbb{R}^{m_y}}{\operatorname{argmin}} \sum_{u=1}^t \sum_{k=1}^{\Phi_u} \left( y_{u,\sigma_{u,k}}^\top \beta - \gamma_{u,k} \right)^2 + \alpha_\Lambda \|\beta\|_2^2 = \left( \Sigma_t^\Lambda \right)^{-1} \rho_t^\Lambda, \tag{12}$$

$$\hat{\beta}_t^Q = \underset{\beta \in \mathbb{R}^{m_x}}{\operatorname{argmin}} \sum_{u=1}^t \sum_{k=1}^{\Phi_u} \mathbb{I}[\gamma_{u,k} = 0] \left( x_u^\top \beta - \eta_{u,k} \right)^2 + \alpha_Q \|\beta\|_2^2 = \left( \Sigma_t^Q \right)^{-1} \rho_t^Q. \tag{13}$$

Here $\Sigma_t^\Lambda = \sum_{u=1}^t \sum_{k=1}^{\Phi_u} y_{u,\sigma_{u,k}} y_{u,\sigma_{u,k}}^\top + \alpha_\Lambda \mathbf{I}_{m_y}$, $\rho_t^\Lambda = \sum_{u=1}^t \sum_{k=1}^{\Phi_u} \gamma_{u,k} y_{u,\sigma_{u,k}}$, $\Sigma_t^Q = \sum_{u=1}^t \sum_{k=1}^{\Phi_u} \mathbb{I}[\gamma_{u,k} = 0] x_u x_u^\top + \alpha_Q \mathbf{I}_{m_x}$, and $\rho_t^Q = \sum_{u=1}^t \sum_{k=1}^{\Phi_u} \mathbb{I}[\gamma_{u,k} = 0] \eta_{u,k} x_u$. $\mathbf{I}_{m_x}$ and $\mathbf{I}_{m_y}$ are the identity matrices with dimension $m_x$ and $m_y$, respectively.

For the estimation of $\beta^S$, similar to Section 5.1, we focus on regressing $w_t = q_t s_t$ instead. Note that $w_t = q_t s_t = x_t^\top \beta^Q (\beta^S)^\top x_t = \operatorname{vec}(x_t x_t^\top)^\top \operatorname{vec}(\beta^Q (\beta^S)^\top)$, where $\operatorname{vec}(\cdot)$ denotes the vectorization operation that maps a matrix to a vector. Therefore, define $z_t \triangleq \operatorname{vec}(x_t x_t^\top)$ and $\beta^W \triangleq \operatorname{vec}(\beta^Q (\beta^S)^\top)$ and we can get $w_t = z_t^\top \beta^W$. Similarly, we estimate $\beta^W$ by conducting ridge regression as follows

$$\hat{\beta}_t^W = \underset{\beta \in \mathbb{R}^{m_x^2}}{\operatorname{argmin}} \sum_{u=1}^t \sum_{k=1}^{\Phi_u} \mathbb{I}[\gamma_{u,k} = 1] \left( z_u^\top \beta - \mu_{u,k} \right)^2 + \alpha_W \|\beta\|_2^2 = \left( \Sigma_t^W \right)^{-1} \rho_t^W. \tag{14}$$

Here $\Sigma_t^W = \sum_{u=1}^t \sum_{k=1}^{\Phi_u} \mathbb{I}[\gamma_{u,k} = 1] z_u z_u^\top + \alpha_W \mathbf{I}_{m_x^2}$ and $\rho_t^W = \sum_{u=1}^t \sum_{k=1}^{\Phi_u} \mathbb{I}[\gamma_{u,k} = 1] \mu_{u,k} z_u$. $\mathbf{I}_{m_x^2}$ is the identity matrix with dimension $m_x^2$.

Let $\|\beta\|_A = \sqrt{\beta^\top A \beta}$ for any positive definite matrix $A$. Proposition 5.4 gives the estimation error bound of the coefficients.

**Proposition 5.4.** *Under Assumptions 5.1 and 5.2 (with parameter U), for any $t \geq 1$, with probability at least $1 - 3t^{-2}$, the following holds:*

$$\left\| \hat{\beta}_t^\Lambda - \beta^\Lambda \right\|_{\Sigma_t^\Lambda} \leq \tau(t, m_y, \alpha_\Lambda), \left\| \hat{\beta}_t^Q - \beta^Q \right\|_{\Sigma_t^Q} \leq \tau(t, m_x, \alpha_Q), and \left\| \hat{\beta}_t^W - \beta^W \right\|_{\Sigma_t^W} \leq \tau(t, m_x^2, \alpha_W). \tag{15}$$

*Here $\tau(t, m, \alpha) = 1/2 \sqrt{m \log\left(1 + tN/(m\alpha)\right) + 4\log t} + \alpha^{1/2} (U + 1)^2$.*

## B)  Algorithm

Given the estimation of $\beta^\Lambda$, $\beta^Q$ and $\beta^W$ we estimate $\lambda_{t,k}$, $q_t$ and $w_t$ as follows.

$$\tilde{\lambda}_{t,k} = \operatorname{Proj}_{[0,1]} \left( y_{t,k}^\top \hat{\beta}_{t-1}^\Lambda + \tau(t-1, m_y, \alpha_\Lambda) \|y_{t,k}\|_{(\Sigma_{t-1}^\Lambda)^{-1}} \right)$$

$$\tilde{q}_t = \operatorname{Proj}_{[0,1-\epsilon_Q]} \left( x_t^\top \hat{\beta}_{t-1}^Q + \tau(t-1, m_x, \alpha_Q) \|x_t\|_{(\Sigma_{t-1}^Q)^{-1}} \right) \tag{16}$$

$$\tilde{w}_t = \operatorname{Proj}_{[0,\tilde{q}_t]} \left( z_t^\top \hat{\beta}_{t-1}^W + \tau(t-1, m_x^2, \alpha_W) \|z_t\|_{(\Sigma_{t-1}^W)^{-1}} \right)$$

Here $\operatorname{Proj}_{[a,b]}(\cdot)$ is a function that projects the input into the interval $[a, b]$.

As shown in Algorithm 2, when the $t$-th consumer arrives, the online retailer observes the consumer features $x_t$ and $y_{t,k}$ for $k \in [N]$ and calculates the intermediate feature $z_t$. Afterward, he calculates $\tilde{\lambda}_{t,k}$, $\tilde{q}_t$, and $\tilde{w}_t$ according to Equation (16) in Line 5. Then the ranking policy $\sigma_t$ is given as shown in Line 6 and the statistics are updated in Lines 7 and 8. Finally, the parameters $\hat{\beta}_t^\Lambda$, $\hat{\beta}_t^Q$, and $\hat{\beta}_t^W$ are estimated in Line 9 for the next iteration.

## C)  Regret Analysis

We analyze the performance of Algorithm 2 by investigating its regret bound. For a sequence of $T$ consumers with features $\boldsymbol{X}_T = \{x_t\}_{t=1}^T$ and $\boldsymbol{Y}_T = \{\{y_{t,k}\}_{k=1}^N\}_{t=1}^T$, the regret is given by

$$\operatorname{Reg}_T(\boldsymbol{X}_T, \boldsymbol{Y}_T) = \sum_{t=1}^T \operatorname{Revenue}(\sigma_t^*; \lambda_t, q_t, s_t) - \operatorname{Revenue}(\sigma_t; \lambda_t, q_t, s_t). \tag{17}$$

---
**Algorithm 2:** MPB-UCB (Contextual)
---
1 **Input:** Products revenue $r$ and hyper-parameter $\epsilon_Q$, $\alpha_\Lambda$, $\alpha_Q$, and $\alpha_W$.
2 **Initialization:** $\hat{\beta}_0^\Lambda = \mathbf{0}$, $\hat{\beta}_0^Q = \mathbf{0}$, and $\hat{\beta}_0^W = \mathbf{0}$.
3 **for** *t=1:T* **do**
4      Observe consumer features $x_t$ and $y_{t,k}$ for $k \in [N]$. Let $z_t = \text{vec}(x_t x_t^\top)$.
5      Calculate $\tilde{\lambda}_{t,k}$, $\tilde{q}_t$, and $\tilde{w}_t$ according to Equation (16).
6      Let $\boldsymbol{\sigma}_t$ be the optimal ranking policy from Theorem 4.1 with $\boldsymbol{\lambda} = \tilde{\boldsymbol{\lambda}}_t$, $q = \tilde{q}_t$, and $s = \tilde{w}_t/\tilde{q}_t$.
7      Offer ranking policy $\boldsymbol{\sigma}_t$ and observe $\Phi_t$, $\boldsymbol{\gamma}_t$, $\boldsymbol{\eta}_t$, $\boldsymbol{\mu}_t$.
8      Calculate statistics $\Sigma_t^\Lambda$, $\rho_t^\Lambda$, $\Sigma_t^Q$, $\rho_t^Q$, $\Sigma_t^W$, and $\rho_t^W$.
9      Calculate estimated parameters $\hat{\beta}_t^\Lambda$, $\hat{\beta}_t^Q$, and $\hat{\beta}_t^W$ according to Equations (12), (13) and (14).
10 **end**
---

Similar to the non-contextual setting, $\text{Reg}_T(\boldsymbol{X}_T, \boldsymbol{Y}_T)$ is a random variable and we focus on $\mathbb{E}[\text{Reg}_T(\boldsymbol{X}_T, \boldsymbol{Y}_T)]$. The performance of the algorithm is given by the following theorem.

**Theorem 5.5.** *Under Assumptions 3.1 (with parameter $\epsilon_Q$), 5.1, and 5.2 (with parameter U), for any $T > 1$, the regret achieved by Algorithm 2 is bounded by*

$$\mathbb{E}[\text{Reg}_T(\boldsymbol{X}_T, \boldsymbol{Y}_T)] \le C \cdot \frac{r_{\max}}{\epsilon_Q} \left( \chi(T, m_y, \alpha_\Lambda) + \chi(T, m_x, \alpha_Q) + \chi(T, m_x^2, \alpha_W) \right). \quad (18)$$

*Here $C$ is a absolute constant (independent of problem parameters), $r_{\max} = \max_{k \in [N]} r_k$, and*

$$\chi(T, m, \alpha) = \left( \frac{1}{2} \sqrt{m \log\left(1 + \frac{TN}{m\alpha}\right) + 4 \log T} + \alpha^{1/2}(U+1)^2 \right) \sqrt{\frac{TN^2 m \log\left(1 + \frac{TN}{m\alpha}\right)}{\alpha \log(1 + \alpha^{-1})}}. \quad (19)$$

**Remark.** We analyze different parameters' impacts on the regret bound in Equation (19). Firstly, the regret grows at $\tilde{O}(\sqrt{T})$, a standard result in online learning algorithms [46]. Secondly, the regret depends on $\tilde{O}(N)$, which shares a similar result with the revenue management literature [17]. Thirdly, similar to Theorem 5.3, the bound depends linearly on $r_{\max}/\epsilon_Q$ that measures the maximal expected revenue the online retailer could achieve from each consumer. Fourthly, the bound depends on the dimensions of features (*i.e.*, $m_x$ and $m_y$) by $\tilde{O}(\sqrt{m_y})$ and $\tilde{O}(m_x)$ because the feature dimension for the estimation of $w_t = q_t s_t$ is $m_x^2$ and then the result is consistent with [17, 54]. Finally, the bound depends on $O(U^2 + U)$ since $\|\beta^W\| \le U^2$ and $\|\beta^Q\|, \|\beta^\Lambda\| \le U$, which agrees with [17].

## 6 Experiments

In this section, we conduct experiments on synthetic data to verify the performances of the online ranking policies. Reports and discussions of more results including the expected revenue and the ratio over the expected revenue of the optimal ranking policy are in Section B.

**Baselines** We adopt the following baselines. Firstly, Cao and Sun [13] considered the single-purchase setting, which is the simplified version of our multiple-purchase setting. We denote it as **Single Purchase**. Secondly, we adopt the method in [40] (denoted as **Keep Viewing**), which assumes that consumers always keep viewing the ranking list after purchasing products. Furthermore, we implement two explore-then-exploit-based algorithms (denoted as **Explore Then Exploit A** and **Explore Then Exploit B**, respectively) to verify the advantages of our method that could balance exploration and exploitation.

### 6.1 Synthetic Data

**Data-generating processes** For the non-contextual setting, we consider $N = 50$ and $300$ products and $T = 100,000$ consumers. The revenue for each product $r_k$ is uniformly distributed in $[0, 1]$ and the purchase probabilities for each product are uniformly sampled from $[0, 0.3]$. We set $q = 0.9$ and $s = 0.5$ and $0.8$ to test the performance of different models on characterizing consumers' behaviors with multiple purchases.

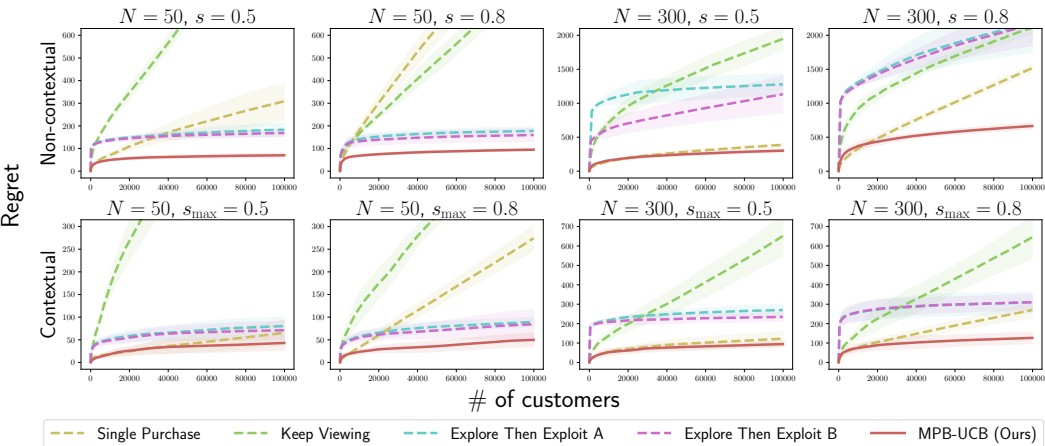

Figure 1: Experimental results on synthetic data. We plot the regret curves for different settings by varying $N = 50/300$ and $s/s_{\max} = 0.5/0.8$ in both non-contextual and contextual scenarios.

For the contextual setting, we consider $N = 50$ and $300$ products and $T = 100,000$ consumers. The revenue $r_k$ is also uniformly distributed in $[0, 1]$. Both the dimension of consumer features and product features are set to 5 (*i.e.*, $m_x = 5$). The joint feature $y_{t,k}$ (the feature of the $t$-th consumer and $k$-th product) is a concatenate of the corresponding consumer and product feature so that $m_y = 10$. The consumer features and product features are uniformly sampled from $[0.8/\sqrt{m_x}, 1/\sqrt{m_x}]$ and $[0, 1/\sqrt{m_x}]$. Afterward, the coefficients $\beta^\Lambda$, $\beta^Q$, and $\beta^S$ in Equation (11) are uniformly sampled from $[0, 1]$. To ensure $\lambda_{t,k}$, $q_t$, $s_t$ have similar ranges in the contextual setting, we normalize the coefficients such that the maximal elements in $\lambda_{t,k}$ and $q_t$ (denoted as $\lambda_{\max}$ and $q_{\max}$) are $0.3$ and $0.9$, respectively. Similarly, by normalizing $\beta^S$, we set the maximal element in $s_t$ (denoted as $s_{\max}$) to $0.5$ and $0.8$.

**Results and analysis** We implement the experiments for 5 distinct simulations by resampling consumers' behaviors while their basic characteristics remain unchanged. Specifically, in both the non-contextual and contextual settings, the purchase probabilities (*i.e.*, $\lambda_t$) and parameters on the span attention and purchase budget (*i.e.*, $q_t$ and $s_t$) are the same. However, the same consumer in distinct simulations may purchase different products due to randomness.

We then evaluate our method (MPB-UCB) and baselines via calculating the regret according to Equations (9) and (17). The results are shown in Figure 1 and our method outperforms all baselines in both settings with various parameters. On the one hand, the Single Purchase and Keep Viewing baselines consider different consumer choice models and are not directly applicable here to the more realistic multiple-purchase setting. On the other hand, compared with the explore-then-exploit-based algorithms, our method integrates the advantages of traditional UCB-like algorithms, leading to a better exploration-exploitation trade-off for smaller regret. In addition, as introduced in Section B, Explore Then Exploit B incorporates learning processes during the exploration phase, leading to better performance compared with Explore Then Exploit A.

### 6.2 Semi-synthetic Data

We utilize the Ad Display/Click Data from Taobao[1], which contains ad display/purchase logs (26 million records) of 1,140,000 anonymized users from the website of Taobao for 8 days. Because the behavior logs only contain the purchase information of the users on a category of a brand, we view a (brand, category) pair in the dataset as a product and the average prices of the ads in this pair as the revenue for the product.

Because we can not obtain consumers' behaviors when we offer a new product ranking policy, following [14], we first estimate the parameters (*i.e.*, $\lambda$, $q$, $s$ in the non-contextual setting and $\beta^\Lambda$, $\beta^Q$, $\beta^S$ in the contextual setting) with all of the data. Afterward, we use the estimated parameters to simulate consumers' behaviors when we provide them with different ranking lists of products.

---

[1] https://tianchi.aliyun.com/datalab/dataSet.html?dataId=56. License: CC BY-NC 4.0.

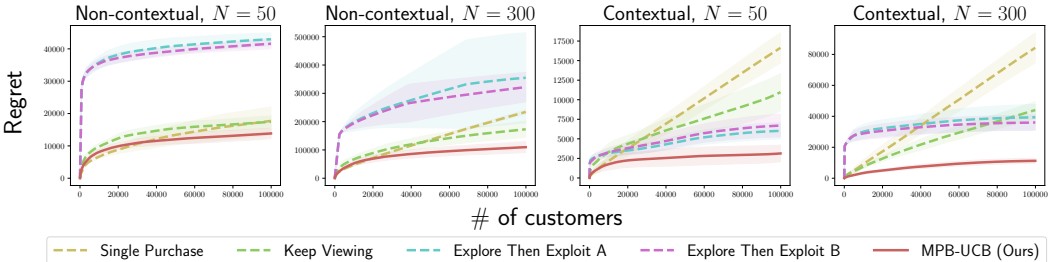

Figure 2: Experimental results on the semi-synthetic data. We plot the regret curves for different settings by varying $N = 50/300$ in both non-contextual and contextual scenarios.

**Data processing** Since each consumer may launch and leave the platform many times during the 8 days, we need to distinguish different launching activities of the same consumer. Specifically, we suppose that a consumer stops viewing the product list if it has been more than 10 minutes since her last interaction with the platform. Afterward, we can estimate the parameters in both non-contextual and contextual settings.

In the non-contextual setting, we calculate the statistics $C_{t,k}$, $c_{t,k}$, $D_t^Q$, $d_t^Q$, $D_t^W$, and $d_t^W$ from the data and estimate $\boldsymbol{\lambda}$, $q$, $s$ directly according to Lemma 5.1. The estimation is $q = 0.870$ and $s = 0.907$. Afterward, we sample $N = 50$ and $300$ products with prices no more than $200$ and purchase probabilities at least $0.1$. Similar to the synthetic data, we set $T = 100,000$.

In the contextual setting, we set the gender, age level, consumption grade, and shopping level as consumers' features and the number of ads, the logarithm of the average, standard deviation, maximum, and median of the ads' prices in the (brand, category) pair as products' features. With the statistics $C_{t,k}$, $c_{t,k}$, $D_t^Q$, $d_t^Q$, $D_t^W$, and $d_t^W$, we use the ridge regression with regularization strength $1$ to estimate $\beta^\Lambda$, $\beta^Q$, and $\beta^S$. Afterward, we sample $N = 50$ and $300$ products with prices no more than $200$ and purchase probabilities at least $0.1$. In addition, we sample $T = 100,000$ consumers.

**Results and analysis** Similar to the synthetic experiments, we implement the experiments for 5 distinct simulations by resampling consumers' behaviors while their basic characteristics remain unchanged. The results of the regret are shown in Figure 2. As shown in the figure, our method outperforms all baselines in different scenarios. Firstly, the Single Purchase and Keep Viewing baselines are not directly applicable here since they consider different consumer choice models. It is worth mentioning that the estimated $s$ is large from the dataset. As a result, the consumers prefer to keep viewing products if they purchase any product, making the Keep Viewing baseline outperforms Single Purchase in this setting. Secondly, our method combines the benefits of existing UCB-like algorithms, resulting in a superior exploration-exploitation trade-off with less regret compared with explore-then-exploit-based methods. In addition, similar to the analysis in the synthetic experiments, Explore Then Exploit B outperforms Explore Then Exploit A in most cases.

## 7 Conclusion

To conclude, we propose a novel consumer choice model to deal with multiple-purchase activities of consumers in online scenarios. We characterize the optimal ranking policy and further design the MPB-UCB algorithms with $\tilde{O}(\sqrt{T})$ regret in both non-contextual and contextual online settings. We conduct extensive experiments to prove the effectiveness of our method.

## Acknowledgments

Peng Cui's research was supported in part by National Key R&D Program of China (No. 2018AAA0102004, No. 2020AAA0106300), National Natural Science Foundation of China (No. U1936219, 62141607), Beijing Academy of Artificial Intelligence (BAAI). Bo Li's research was supported by the National Natural Science Foundation of China (No.72171131), the Tsinghua University Initiative Scientific Research Grant (No. 2019THZWJC11), Technology and Innovation Major Project of the Ministry of Science and Technology of China under Grants 2020AAA0108400 and 2020AAA0108403.

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
