## A  Limitations and Societal Impacts

One potential limitation is that we suppose the distribution of the attention span and purchase budget obey the geometric distribution. While the assumption is common [13, 40], we can relax it in future work, similar to [17, 14].

This paper does not have critical negative societal impacts because the purpose of the paper is to design an optimal ranking policy for online retailers. Besides the online retailers, consumers may also benefit because Equation (5) indicates that a product with a higher purchase probability may appear in a higher order in the ranking list. As a result, consumers may find the products shown in the ranking list highly relevant.

## B  More Experimental Results

### B.1  Implementation Details

**Baselines**

• **Single Purchase** [13]. Cao and Sun [13] suppose that consumers always purchase at most one product. As a result, we implement them by setting $\tilde{s}_t = 0$ for all consumers. The estimations of $\tilde{\lambda}_{t,k}$ and $\tilde{q}_t$ are the same as our method.
• **Keep Viewing** [40]. Liang et al. [40] suppose that consumers always keep viewing the list if they purchase one product. To get the optimal ranking policy with the knowledge of consumers' characteristics, we can sort the products with value $(\lambda_k r_k)/(1 - q - (1 - q)\lambda_k)$ [40, Theorem 1]. Afterward, we use similar techniques to estimate $\tilde{\lambda}_{t,k}$ and $\tilde{q}_t$ and maximize the revenue in the online settings.
• **Explore Then Exploit A**. Following [13], we implement two explore-then-exploit-based algorithms as baselines. For the Explore Then Exploit A method, we first adopt an exploration stage to ensure that each product is displayed to consumers at least $\delta \log T$ times. (Here $\delta$ is a hyper-parameter.) To be specific, in the exploration stage, we rank the products according to their current numbers of displays. Afterward, we use the estimated parameters to provide the optimal ranking policy according to Theorem 4.1 in the main body of the paper.
• **Explore Then Exploit B**. This method is a variant of the Explore Then Exploit A method. In detail, in the exploration stage, after ranking the products according to their number of displays, we further rank the products that have been displayed to consumers at least $\delta \log T$ times according to Theorem 4.1 in the main body.

**Hyper-parameters**  We add hyper-parameters $\xi_\Lambda$, $\xi_Q$, and $\xi_W$ to determine the level of exploration for our method and two baselines (Single purchase and Keep viewing). The other two baselines do not include the hyper-parameter $\xi$ because they do not have the dynamic exploration step. In detail, in the non-contextual setting, the parameters $\tilde{\lambda}_{t,k}$, $\tilde{q}_t$, and $\tilde{w}_t$ of our method are calculated as

$$\tilde{\lambda}_{t,k} \triangleq \min\left\{1, \hat{\lambda}_{t,k} + \xi_\Lambda\sqrt{\frac{\log t}{C_{t,k}}}\right\}, \tilde{q}_t \triangleq \min\left\{1 - \epsilon_Q, \hat{q}_t + \xi_Q\sqrt{\frac{\log t}{D_t^Q}}\right\}, \tilde{w}_t \triangleq \min\left\{\tilde{q}_t, \hat{w}_t + \xi_W\sqrt{\frac{\log t}{D_t^W}}\right\}.$$
(20)

In the contextual setting, the parameters are also calculated according to the hyper-parameters $\xi_\Lambda$, $\xi_Q$, and $\xi_W$,

$$\tilde{\lambda}_{t,k} = \text{Proj}_{[0,1]}\left(y_{t,k}^\top \hat{\beta}_{t-1}^\Lambda + \xi_\Lambda \cdot \tau(t-1, m_y, \alpha_\Lambda) \|y_{t,k}\|_{(\Sigma_{t-1}^\Lambda)^{-1}}\right),$$

$$\tilde{q}_t = \text{Proj}_{[0,1-\epsilon_Q]}\left(x_t^\top \hat{\beta}_{t-1}^Q + \xi_Q \cdot \tau(t-1, m_x, \alpha_Q) \|x_t\|_{(\Sigma_{t-1}^Q)^{-1}}\right),$$
(21)

$$\tilde{w}_t = \text{Proj}_{[0,\tilde{q}_t]}\left(z_t^\top \hat{\beta}_{t-1}^W + \xi_W \cdot \tau(t-1, m_x^2, \alpha_W) \|z_t\|_{(\Sigma_{t-1}^W)^{-1}}\right).$$

For the synthetic dataset, we search the hyper-parameters $\xi_\Lambda \in \{0.1, 0.3, 0.5\}$, $\xi_Q, \xi_W \in \{0.05, 0.1, 0.2\}$, $\delta \in \{0.5, 1, 2, 5, 10\}$, and $\alpha_Q = \alpha_W = \alpha_\Lambda \in \{0.01, 0.1, 1\}$. For the semi-synthetic dataset, we search the hyper-parameters $\xi_\Lambda \in \{0.1, 0.3, 0.5\}$, $\xi_Q, \xi_W \in \{0.05, 0.1, 0.2\}$,

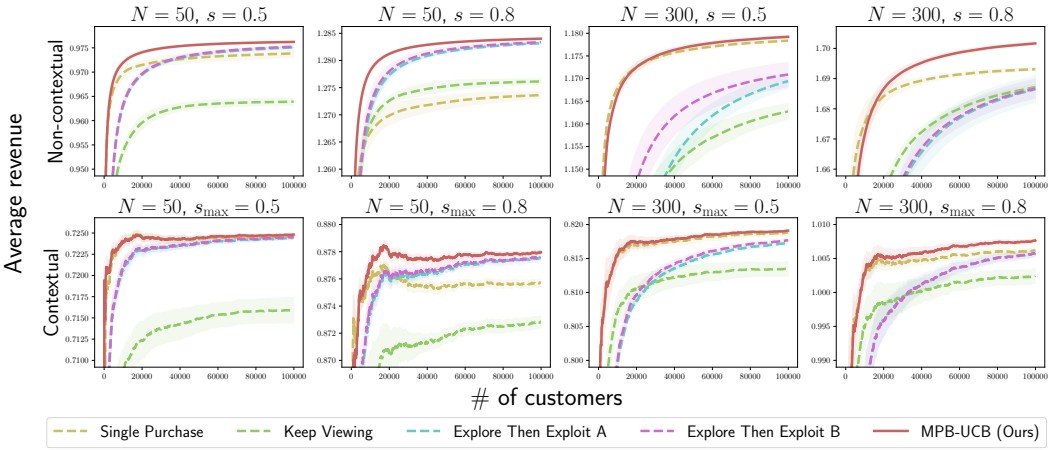

Figure 3: The average revenue metric on the synthetic data.

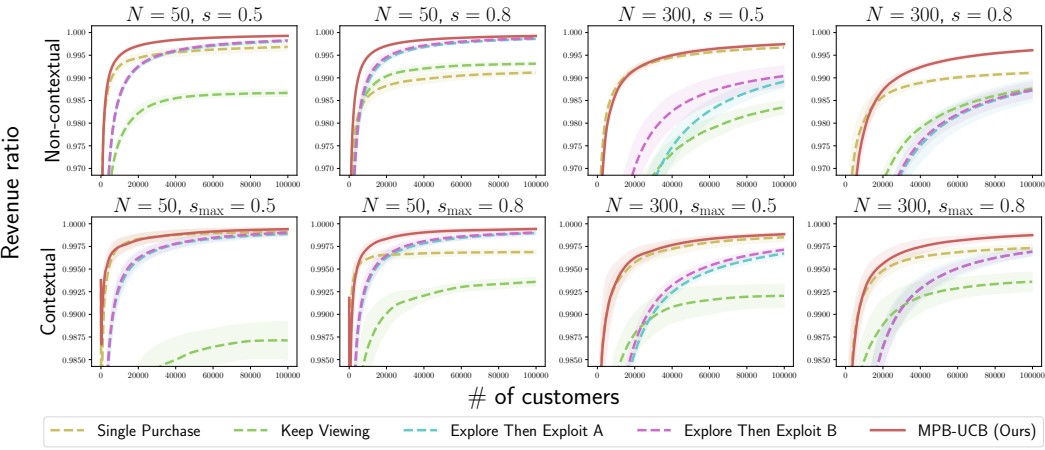

Figure 4: The revenue ratio metric on the synthetic data.

$\delta \in \{0.5, 1, 2, 5, 10\}$, and $\alpha_Q = \alpha_W = \alpha_\Lambda \in \{0.1, 1, 10\}$. The grid search code is based on the NNI package[2].

## B.2  More Results on the Synthetic and Semi-synthetic Data

We further report two more metrics to verify the effectiveness of our method. In detail, Average revenue($T$) measures the average revenue achieved by different algorithms till the $T$-th consumer. Revenue ratio($T$) measures the ratio of the achieved revenue over the optimal ranking policy. Formally, the metrics are calculated as follows.

$$
\begin{aligned}
\text{Average revenue}(T) &= \frac{1}{T} \sum_{t=1}^{T} \text{Revenue}(\boldsymbol{\sigma}_t; \boldsymbol{\lambda}_t, q_t, s_t), \\
\text{Revenue ratio}(T) &= \frac{\sum_{t=1}^{T} \text{Revenue}(\boldsymbol{\sigma}_t; \boldsymbol{\lambda}_t, q_t, s_t)}{\sum_{t=1}^{T} \text{Revenue}(\boldsymbol{\sigma}_t^*; \boldsymbol{\lambda}_t, q_t, s_t)}.
\end{aligned}
\tag{22}
$$

For both metrics, higher values indicate better performances. The results of these two metrics on both the synthetic and Semi-synthetic datasets are shown in Figures 3, 4, 5, and 6. These figures further demonstrate that our method outperforms all baselines in both contextual and non-contextual settings with various parameters (*i.e.*, $s$, $s_{\max}$, and $N$).

---

[2]https://github.com/microsoft/nni. Licence: MIT License.

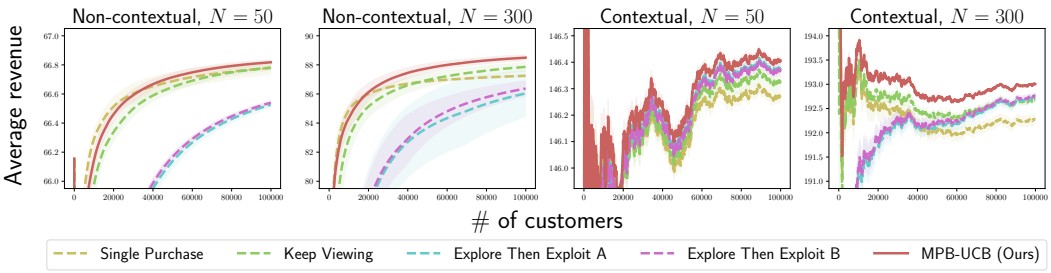

Figure 5: The average revenue metric on the semi-synthetic data.

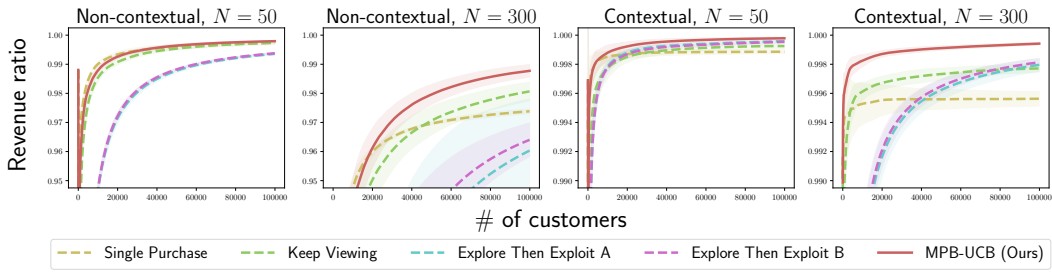

Figure 6: The revenue ratio metric on the semi-synthetic data.

## C Omitted Proofs

### C.1 Proof of Theorem 4.1

*Proof.* We drop all the subscript $t$ for simplicity. We first write the expected revenue function.

$$
\begin{aligned}
&\text{Revenue}\,(\boldsymbol{\sigma}; \boldsymbol{\lambda}, q, s) \\
&= \mathbb{E}_{V \sim \text{Geo}(1-q), B \sim \text{Geo}(1-s)} \left[ R(V, B; \boldsymbol{\lambda}, \boldsymbol{\sigma}) \right] \\
&= \sum_{v=1}^{N} \sum_{b=1}^{N} P(V=v) P(B=b) \left( \sum_{Q \subseteq [v], |Q| \leq b} P^{\text{buy}}(Q; v, b, \boldsymbol{\lambda}, \boldsymbol{\sigma}) \left( \sum_{k \in Q} r_{\sigma_k} \right) \right) \\
&= \sum_{k=1}^{N} r_{\sigma_k} \left( \sum_{v=1}^{N} \sum_{b=1}^{N} P(V=v) P(B=b) \left( \sum_{Q \subseteq [N], k \in Q} P^{\text{buy}}(Q; v, b, \boldsymbol{\lambda}, \boldsymbol{\sigma}) \right) \right) \\
&= \sum_{k=1}^{N} r_{\sigma_k} \left( \sum_{v=k}^{N} \sum_{b=1}^{N} P(V=v) P(B=b) \left( \sum_{Q \subseteq [N], k \in Q} P^{\text{buy}}(Q; v, b, \boldsymbol{\lambda}, \boldsymbol{\sigma}) \right) \right).
\end{aligned}
\tag{23}
$$

For any $Q_k \subseteq [N]$ that satisfies $k \in Q_k$, define $Q_k^- \triangleq \{i \in Q_k : i < k\}$ and $Q_k^+ \triangleq \{i \in Q_k : i > k\}$. We further define

$$
G_k^-(Q_k^-) \triangleq \left( \prod_{i \in Q_k^-} \lambda_{\sigma_i} \right) \left( \prod_{1 \leq i < k, i \notin Q_k^-} (1 - \lambda_{\sigma_i}) \right).
\tag{24}
$$

As a result,

$$\sum_{v=k}^{N}\sum_{b=1}^{N}P(V=v)P(B=b)\left(\sum_{Q\subseteq[N],k\in Q}P^{\mathrm{buy}}(Q;v,b,\boldsymbol{\lambda},\boldsymbol{\sigma})\right)$$

$$=\sum_{v=k}^{N}\sum_{b=1}^{N}P(V=v)P(B=b)\left(\sum_{Q_k^-\subseteq[k-1]}\sum_{Q_k^+\subseteq[N]\backslash[N]}P^{\mathrm{buy}}(Q;v,b,\boldsymbol{\lambda},\boldsymbol{\sigma})\right)$$

$$=\lambda_{\sigma_k}\sum_{v=k}^{N}\sum_{b=1}^{N}P(V=v)P(B=b)\left(\sum_{Q_k^-\subseteq[k-1]}\sum_{Q_k^+\subseteq[N]\backslash[N]}G_k^-(Q_k^-)P_{k+1}^{\mathrm{buy}}\left(Q_k^+;v,b-|Q_k^-|-1,\boldsymbol{\lambda},\boldsymbol{\sigma}\right)\right).$$
(25)

Here $P_{k+1}^{\mathrm{buy}}\left(Q_k^+;v,b-|Q_k^-|-1,\boldsymbol{\lambda},\boldsymbol{\sigma}\right)$ in the second equality denotes the probability of purchasing $Q_k^+$ with attention span $v$ and purchase budget $b$ after purchasing $|Q_k^-|+1$ products, *i.e.*,

$$P_{k+1}^{\mathrm{buy}}\left(Q_k^+;v,b-|Q_k^-|-1,\boldsymbol{\lambda},\boldsymbol{\sigma}\right)=\begin{cases}\left(\prod_{i\in Q_k^+}\lambda_{\sigma_i}\right)\left(\prod_{k<i\leq v}(1-\lambda_{\sigma_i})\right), & \text{if }|Q_k^+|<b-|Q_k^-|-1.\\\left(\prod_{i\in Q_k^+}\lambda_{\sigma_i}\right)\left(\prod_{k<i\leq\max(Q_k^+)}(1-\lambda_{\sigma_i})\right), & \text{if }|Q_k^+|=b-|Q_k^-|-1.\\0, & \text{if }|Q_k^+|>b-|Q_k^-|-1.\end{cases}$$
(26)

As a result, when $|Q_k^-|$, $v$, $b$, $\boldsymbol{\lambda}$, and $\boldsymbol{\sigma}$ are fixed, by definition, we have

$$\sum_{Q_k^+\subseteq[N]\backslash[N]}P_{k+1}^{\mathrm{buy}}\left(Q_k^+;v,b-|Q_k^-|-1,\boldsymbol{\lambda},\boldsymbol{\sigma}\right)=1$$
(27)

Hence,

$$\lambda_{\sigma_k}\sum_{v=k}^{N}\sum_{b=1}^{N}P(V=v)P(B=b)\left(\sum_{Q_k^-\subseteq[k-1]}\sum_{Q_k^+\subseteq[N]\backslash[N]}G_k^-(Q_k^-)P_{k+1}^{\mathrm{buy}}\left(Q_k^+;v,b-|Q_k^-|-1,\boldsymbol{\lambda},\boldsymbol{\sigma}\right)\right)$$

$$=\lambda_{\sigma_k}\sum_{v=k}^{N}\sum_{b=1}^{N}P(V=v)P(B=b)\left(\sum_{u=0}^{b-1}\left(\sum_{Q_k^-\subseteq[k-1],|Q_k^-|=u}G_k^-(Q_k^-)\right)\left(\sum_{Q_k^+\subseteq[N]\backslash[N]}P_{k+1}^{\mathrm{buy}}\left(Q_k^+;v,b-u-1,\boldsymbol{\lambda},\boldsymbol{\sigma}\right)\right)\right)$$

$$=\lambda_{\sigma_k}\sum_{v=k}^{N}\sum_{b=1}^{N}P(V=v)P(B=b)\left(\sum_{u=0}^{b-1}\left(\sum_{Q_k^-\subseteq[k-1],|Q_k^-|=u}G_k^-(Q_k^-)\right)\right)$$

$$=\lambda_{\sigma_k}P(V\geq k)\sum_{b=1}^{N}P(B=b)\left(\sum_{u=0}^{b-1}\left(\sum_{Q_k^-\subseteq[k-1],|Q_k^-|=u}G_k^-(Q_k^-)\right)\right)$$

$$=\lambda_{\sigma_k}P(V\geq k)\sum_{u=0}^{k-1}P(B\geq u+1)\left(\sum_{Q_k^-\subseteq[k-1],|Q_k^-|=u}G_k^-(Q_k^-)\right).$$
(28)

For any $k\geq 1$ and $0\leq u\leq k-1$, define

$$H_{\boldsymbol{\sigma}}(k,u;\boldsymbol{\lambda})\triangleq\sum_{Q_k^-\subseteq[k-1],|Q_k^-|=u}G_k^-(Q_k^-).$$
(29)

Intuitively, $H_{\boldsymbol{\sigma}}(k,u;\boldsymbol{\lambda})$ measures the probability that a consumer purchases exactly $u$ products from the position index 1 to $k-1$ if ignoring the attention span and purchase budget. As a result,

$$\mathrm{Revenue}\left(\boldsymbol{\sigma};\boldsymbol{\lambda},q,s\right)=\sum_{k=1}^{N}r_{\sigma_k}\lambda_{\sigma_k}P(V\geq k)\left(\sum_{u=0}^{k-1}P(B\geq u+1)H_{\boldsymbol{\sigma}}(k,u;\boldsymbol{\lambda})\right)$$

$$=\sum_{k=1}^{N}r_{\sigma_k}\lambda_{\sigma_k}q^{k-1}\left(\sum_{u=0}^{k-1}s^uH_{\boldsymbol{\sigma}}(k,u;\boldsymbol{\lambda})\right).$$
(30)

In addition, it is easy to verify that $H_{\boldsymbol{\sigma}}(k, u; \boldsymbol{\lambda})$ has the following recurrence relation

$$H_{\boldsymbol{\sigma}}(1, 0; \boldsymbol{\lambda}) = 1.$$

$$H_{\boldsymbol{\sigma}}(k, 0; \boldsymbol{\lambda}) = \prod_{i=1}^{k-1}(1 - \lambda_{\sigma_i}), \quad H_{\boldsymbol{\sigma}}(k, k-1; \boldsymbol{\lambda}) = \prod_{i=1}^{k-1} \lambda_{\sigma_i}, \qquad\qquad \forall k \geq 2.$$

$$H_{\boldsymbol{\sigma}}(k, u; \boldsymbol{\lambda}) = \left(1 - \lambda_{\sigma_{k-1}}\right) H_{\boldsymbol{\sigma}}(k-1, u; \boldsymbol{\lambda}) + \lambda_{\sigma_{k-1}} H_{\boldsymbol{\sigma}}(k-1, u-1; \boldsymbol{\lambda}), \quad \forall k \geq 2, 0 < u < k-1. \tag{31}$$

From the definition of $G_k^-$ and $H_{\boldsymbol{\sigma}}$, it is easy to check that the value of $H_{\boldsymbol{\sigma}}(k, u; \boldsymbol{\lambda})$ does not depend on the order of $\sigma_1, \sigma_2, \ldots, \sigma_{k-1}$. As a result, for any $1 \leq i < N$, let $\boldsymbol{\sigma}$ be a ranking policy that differs from $\boldsymbol{\sigma}^*$ only in the $i$-th and $(i+1)$-th position, $i.e.$,

$$\sigma_j = \begin{cases} \sigma_{i+1}^*, & \text{if } j = i. \\ \sigma_i^*, & \text{if } j = i+1. \\ \sigma_j^*, & \text{otherwise.} \end{cases} \tag{32}$$

Due to the optimality of $\boldsymbol{\sigma}^*$, we have

$$0 \leq \text{Revenue}\,(\boldsymbol{\sigma}^*; \boldsymbol{\lambda}, q, s) - \text{Revenue}\,(\boldsymbol{\sigma}; \boldsymbol{\lambda}, q, s)$$

$$= r_{\sigma_i^*} \lambda_{\sigma_i^*} q^{i-1} \left( \sum_{u=0}^{i-1} s^u H_{\boldsymbol{\sigma}^*}(i, u; \boldsymbol{\lambda}) \right) + r_{\sigma_{i+1}^*} \lambda_{\sigma_{i+1}^*} q^i \left( \sum_{u=0}^{i} s^u H_{\boldsymbol{\sigma}^*}(i+1, u; \boldsymbol{\lambda}) \right)$$

$$- r_{\sigma_i} \lambda_{\sigma_i} q^{i-1} \left( \sum_{u=0}^{i-1} s^u H_{\boldsymbol{\sigma}}(i, u; \boldsymbol{\lambda}) \right) + r_{\sigma_{i+1}} \lambda_{\sigma_{i+1}} q^i \left( \sum_{u=0}^{i} s^u H_{\boldsymbol{\sigma}}(i+1, u; \boldsymbol{\lambda}) \right)$$

$$= r_{\sigma_i^*} \lambda_{\sigma_i^*} q^{i-1} \left( \sum_{u=0}^{i-1} s^u H_{\boldsymbol{\sigma}^*}(i, u; \boldsymbol{\lambda}) \right) - r_{\sigma_i} \lambda_{\sigma_i} q^{i-1} \left( \sum_{u=0}^{i-1} s^u H_{\boldsymbol{\sigma}}(i, u; \boldsymbol{\lambda}) \right)$$

$$+ r_{\sigma_{i+1}^*} \lambda_{\sigma_{i+1}^*} q^i \left( \left(1 - \lambda_{\sigma_i^*}\right) \sum_{u=0}^{i-1} s^u H_{\boldsymbol{\sigma}^*}(i, u; \boldsymbol{\lambda}) + \lambda_{\sigma_i^*} \sum_{u=1}^{i} s^u H_{\boldsymbol{\sigma}^*}(i, u-1; \boldsymbol{\lambda}) \right) \tag{33}$$

$$- r_{\sigma_{i+1}} \lambda_{\sigma_{i+1}} q^i \left( \left(1 - \lambda_{\sigma_i}\right) \sum_{u=0}^{i-1} s^u H_{\boldsymbol{\sigma}}(i, u; \boldsymbol{\lambda}) + \lambda_{\sigma_i} \sum_{u=1}^{i} s^u H_{\boldsymbol{\sigma}}(i, u-1; \boldsymbol{\lambda}) \right)$$

$$= q^{i-1} \left( \sum_{u=0}^{i-1} s^u H_{\boldsymbol{\sigma}}(i, u; \boldsymbol{\lambda}) \right) \cdot$$

$$\left( r_{\sigma_i^*} \lambda_{\sigma_i^*} \left(1 - q + q(1-s)\lambda_{\sigma_{i+1}^*}\right) - r_{\sigma_{i+1}^*} \lambda_{\sigma_{i+1}^*} \left(1 - q + q(1-s)\lambda_{\sigma_i^*}\right) \right).$$

Hence,

$$\frac{r_{\sigma_i^*} \lambda_{\sigma_i^*}}{1 - q + q(1-s)\lambda_{\sigma_i^*}} \geq \frac{r_{\sigma_{i+1}^*} \lambda_{\sigma_{i+1}^*}}{1 - q + q(1-s)\lambda_{\sigma_{i+1}^*}}. \tag{34}$$

Now the claim follows. □

## C.2 Proof of Lemma 5.1

*Proof.* Random variables $\gamma_{t,k}$ are independent of each other by the assumptions that the behaviors of different consumers are independent and consumer interests in any two products are independent. For $\eta_{t,k}$, suppose the consumer purchases $u$ products after viewing the $k$-th product, then

$$P\,(\eta_{t,k} = 1) = P\,(V_t > k, B_t > u \mid V_t \geq k, B_t > u)$$

$$= \frac{P\,(V_t > k, B_t > u)}{P\,(V_t \geq k, B_t > u)} = \frac{P\,(V_t > k)\,P\,(B_t > u)}{P\,(V_t \geq k)\,P\,(B_t > u)} = \frac{q_t^k}{q_t^{k-1}} = q_t. \tag{35}$$

As a result, $P\,(\eta_{t,k} = 1)$ does not depend on the historical purchase activities and is always $q_t$. By the assumption that the behaviors of different consumers are independent, $\eta_{t,k}$ are independent Bernoulli random variables and $\eta_{t,k} \sim \text{Ber}(q_t)$.

Similarly, for $\mu_{t,k}$, suppose the consumer purchases $u$ products after viewing the $k$-th product, then

$$
\begin{aligned}
P\left(\mu_{t,k}=1\right) &= P\left(V_t > k, B_t > u \mid V_t \geq k, B_t \geq u\right) \\
&= \frac{P\left(V_t > k, B_t > u\right)}{P\left(V_t \geq k, B_t \geq u\right)} = \frac{P\left(V_t > k\right) P\left(B_t > u\right)}{P\left(V_t \geq k\right) P\left(B_t \geq u\right)} = \frac{q_t^k s_t^u}{q_t^{k-1} s_t^{u-1}} = q_t s_t.
\end{aligned}
\tag{36}
$$

As a result, $P\left(\eta_{t,k}=1\right)$ does not depend on the historical purchase activities. Hence, by the assumption that the behaviors of different consumers are independent, $\mu_{t,k}$ are independent Bernoulli random variables and $\mu_{t,k} \sim \text{Ber}(q_t s_t)$. $\qquad\square$

## C.3 Proof of Proposition 5.2

*Proof.* For each timestamp $t$, define the favorable event as

$$
\xi_t \triangleq \left\{ |q - \hat{q}_t| \leq \sqrt{\frac{2\log t}{D_t^Q}}, |qs - \hat{w}_t| \leq \sqrt{\frac{2\log t}{D_t^W}}, \left|\lambda_k - \hat{\lambda}_{t,k}\right| \leq \sqrt{\frac{2\log t}{C_{t,k}}}, \forall k \in [N] \right\}.
\tag{37}
$$

Let $\bar{\xi}_t$ denote the complement event of $\xi_t$ and $P(\xi_t) = 1 - P(\bar{\xi}_t)$. To estimate $P\left(\bar{\xi}_t\right)$, we need some further notations. Let $\xi_t^Q$ denote the event $\left\{ |q - \hat{q}_t| > \sqrt{2\log t / D_t^Q} \right\}$, $\xi_t^W$ denote the event $\left\{ |qs - \hat{w}_t| > \sqrt{2\log t / D_t^W} \right\}$, and $\xi_t^\Lambda$ denote the event $\left\{ \exists k \in [N], \left|\lambda_k - \hat{\lambda}_{t,k}\right| > \sqrt{2\log t / C_{t,k}} \right\}$. As a result, $\bar{\xi}_t = \xi_t^Q \cup \xi_t^W \cup \xi_t^\Lambda$ and

$$
P(\bar{\xi}_t) \leq P(\xi_t^Q) + P(\xi_t^W) + P(\xi_t^\Lambda)
\tag{38}
$$

by the union bound. We first estimate $P\left(\xi_t^Q\right)$. Let $q_s'$ be the $s$-th observation of $q$ and $\hat{q}_s' = \sum_{u=1}^{s} q_u'/s$ be the average of the top $s$ observations. According to Lemma 5.1 and Hoeffding's inequality, we can get that for any $s \in [Nt]$,

$$
P\left( |q - \hat{q}_s'| > \sqrt{\frac{2\log t}{s}} \right) \leq 2t^{-4}.
\tag{39}
$$

As a result, by the union bound,

$$
\begin{aligned}
P\left(\xi_t^Q\right) = P\left( \left|q - \hat{q}_{D_t^Q}'\right| > \sqrt{\frac{2\log t}{D_t^Q}} \right) &\leq \sum_{s=1}^{tN} P\left( \left|q - \hat{q}_{D_t^Q}'\right| > \sqrt{\frac{2\log t}{D_t^Q}}, D_t^Q = s \right) \\
&\leq \sum_{s=1}^{tN} P\left( |q - \hat{q}_s'| > \sqrt{\frac{2\log t}{s}} \right) \leq \sum_{s=1}^{tN} 2t^{-4} = 2Nt^{-3}.
\end{aligned}
\tag{40}
$$

Using similar techniques, we can get that

$$
P\left(\xi_t^W\right) \leq 2Nt^{-3} \quad \text{and} \quad P\left(\xi_t^\Lambda\right) \leq 2Nt^{-3}.
\tag{41}
$$

Now the claim follows by combining Equations (38), (40), (41). $\qquad\square$

## C.4 Proof of Theorem 5.3

We first define

$$
O_{t,k}^\Lambda \triangleq \mathbb{I}\left[k \leq \Phi_t\right], \quad O_{t,k}^Q \triangleq \mathbb{I}\left[k \leq \Phi_t, \gamma_{t,k} = 0\right], \quad O_{t,k}^W \triangleq \mathbb{I}\left[k \leq \Phi_t, \gamma_{t,k} = 1\right].
\tag{42}
$$

We also need the following two propositions. The subscript $t$ are omitted in the two propositions for simplicity.

**Proposition C.1.** *Suppose $\tilde{\boldsymbol{\lambda}} \succeq \boldsymbol{\lambda}$, $\tilde{q} \geq q$, and $\tilde{s}\tilde{q} \geq sq$, then*

$$
\text{Revenue}\left(\boldsymbol{\sigma}^*; \boldsymbol{\lambda}, q, s\right) \leq \text{Revenue}\left(\boldsymbol{\sigma}^*; \tilde{\boldsymbol{\lambda}}, \tilde{q}, \tilde{s}\right).
\tag{43}
$$

*Proof of Proposition C.1.* For any $\boldsymbol{\sigma} \dashv [N]$, $1 \leq n \leq k$, $0 \leq u \leq k-n$, define

$$J_{\boldsymbol{\sigma}}(n, k, u; \boldsymbol{\lambda}) \triangleq \sum_{Q \subseteq [k-1] \setminus [n-1], |Q|=u} \left( \prod_{i \in Q} \lambda_{\sigma_i} \right) \left( \prod_{n \leq i < k, i \notin Q} \lambda_{\sigma_i} \right). \tag{44}$$

Intuitively, $J_{\boldsymbol{\sigma}}(n, k, u; \boldsymbol{\lambda})$ measures the probability that a consumer purchases exactly $u$ products from the position index $n$ to $k-1$ if ignoring the attention span and purchase budget. As a result, $J_{\boldsymbol{\sigma}}(1, k, u; \boldsymbol{\lambda}) = H_{\boldsymbol{\sigma}}(k, u; \boldsymbol{\lambda})$. In addition, it is easy to check that

$$J_{\boldsymbol{\sigma}}(n, n, 0; \boldsymbol{\lambda}) = 1, \quad \forall 1 \leq n \leq N.$$

$$J_{\boldsymbol{\sigma}}(n-1, k, u; \boldsymbol{\lambda}) = \begin{cases} (1 - \lambda_{\sigma_{n-1}}) J_{\boldsymbol{\sigma}}(n, k, u; \boldsymbol{\lambda}), & \text{if } u = 0. \\ \lambda_{\sigma_{n-1}} J_{\boldsymbol{\sigma}}(n, k, u-1; \boldsymbol{\lambda}), & \text{if } u = k-n+1. \\ (1 - \lambda_{\sigma_{n-1}}) J_{\boldsymbol{\sigma}}(n, k, u; \boldsymbol{\lambda}) + \lambda_{\sigma_{n-1}} J_{\boldsymbol{\sigma}}(n, k, u-1; \boldsymbol{\lambda}), & \text{if } 0 < u < k-n+1. \end{cases} \tag{45}$$

In addition, for any $n \geq 1$, define

$$\tilde{R}_{\boldsymbol{\sigma}}(n; \boldsymbol{\lambda}, q, s) \triangleq \sum_{k=n}^{N} r_{\sigma_k} \lambda_{\sigma_k} q^{k-n} \left( \sum_{u=0}^{k-n} s^u J_{\boldsymbol{\sigma}}(n, k, u; \boldsymbol{\lambda}) \right). \tag{46}$$

Intuitively, $\tilde{R}_{\boldsymbol{\sigma}}(n; \boldsymbol{\lambda}, q, s)$ measures the expected revenue if the top $n-1$ products in the ranking list are not purchased. It is easy to verify that $\tilde{R}_{\boldsymbol{\sigma}}(1; \boldsymbol{\lambda}, q, s) = \text{Revenue}(\boldsymbol{\sigma}; \boldsymbol{\lambda}, q, s)$. Furthermore,

$$\tilde{R}_{\boldsymbol{\sigma}}(n-1; \boldsymbol{\lambda}, q, s)$$

$$= \sum_{k=n-1}^{N} r_{\sigma_k} \lambda_{\sigma_k} q^{k-n+1} \left( \sum_{u=0}^{k-n+1} s^u J_{\boldsymbol{\sigma}}(n-1, k, u; \boldsymbol{\lambda}) \right)$$

$$= r_{\sigma_{n-1}} \lambda_{\sigma_{n-1}} + \sum_{k=n}^{N} r_{\sigma_k} \lambda_{\sigma_k} q^{k-n+1} \left( \sum_{u=0}^{k-n+1} s^u J_{\boldsymbol{\sigma}}(n-1, k, u; \boldsymbol{\lambda}) \right)$$

$$= r_{\sigma_{n-1}} \lambda_{\sigma_{n-1}} + \sum_{k=n}^{N} r_{\sigma_k} \lambda_{\sigma_k} q^{k-n+1} \left( \sum_{u=0}^{k-n} s^u (1 - \lambda_{\sigma_{n-1}}) J_{\boldsymbol{\sigma}}(n, k, u; \boldsymbol{\lambda}) + \sum_{u=1}^{k-n+1} s^u \lambda_{\sigma_{n-1}} J_{\boldsymbol{\sigma}}(n, k, u-1; \boldsymbol{\lambda}) \right)$$

$$= r_{\sigma_{n-1}} \lambda_{\sigma_{n-1}} + q \left( 1 - (1-s) \lambda_{\sigma_{n-1}} \right) \left( \sum_{k=n}^{N} r_{\sigma_k} \lambda_{\sigma_k} q^{k-n} \left( \sum_{u=0}^{k-n} s^u J_{\boldsymbol{\sigma}}(n, k, u; \boldsymbol{\lambda}) \right) \right)$$

$$= r_{\sigma_{n-1}} \lambda_{\sigma_{n-1}} + \left( q - q(1-s) \lambda_{\sigma_{n-1}} \right) \tilde{R}_{\boldsymbol{\sigma}}(n; \boldsymbol{\lambda}, q, s). \tag{47}$$

Now we prove $\tilde{R}_{\boldsymbol{\sigma}^*}(n; \boldsymbol{\lambda}, q, s) \leq \tilde{R}_{\boldsymbol{\sigma}^*}(n; \tilde{\boldsymbol{\lambda}}, \tilde{q}, \tilde{s})$, $\forall n \in [N]$ by induction. Firstly, it is easy to check that $\tilde{R}_{\boldsymbol{\sigma}^*}(N; \boldsymbol{\lambda}, q, s) \leq \tilde{R}_{\boldsymbol{\sigma}^*}(N; \tilde{\boldsymbol{\lambda}}, \tilde{q}, \tilde{s})$. Suppose the condition holds for $\tilde{R}_{\boldsymbol{\sigma}^*}(n; \boldsymbol{\lambda}, q, s) \leq \tilde{R}_{\boldsymbol{\sigma}^*}(n; \tilde{\boldsymbol{\lambda}}, \tilde{q}, \tilde{s})$. Consider the case for $n-1$. Notice that $\tilde{R}_{\boldsymbol{\sigma}^*}(n-1; \boldsymbol{\lambda}, q, s) \geq \tilde{R}_{\boldsymbol{\sigma}^*}(n; \boldsymbol{\lambda}, q, s)$ (otherwise it will increase the revenue if removing the $(n-1)$-th product in the list), we can get that

$$\left( r_{\sigma_{n-1}} - q(1-s) \tilde{R}_{\boldsymbol{\sigma}^*}(n; \boldsymbol{\lambda}, q, s) \right) \lambda_{\sigma_{n-1}} = (1-q) \tilde{R}_{\boldsymbol{\sigma}^*}(n; \boldsymbol{\lambda}, q, s) \geq 0. \tag{48}$$

Hence $r_{\sigma_{n-1}} - q(1-s) \tilde{R}_{\boldsymbol{\sigma}^*}(n; \boldsymbol{\lambda}, q, s) \geq 0$. As a result,

$$\tilde{R}_{\boldsymbol{\sigma}^*}(n-1; \boldsymbol{\lambda}, q, s)$$

$$= r_{\sigma_{n-1}} \lambda_{\sigma_{n-1}} + \left( q - q(1-s) \lambda_{\sigma_{n-1}} \right) \tilde{R}_{\boldsymbol{\sigma}^*}(n; \boldsymbol{\lambda}, q, s)$$

$$= q \tilde{R}_{\boldsymbol{\sigma}^*}(n; \boldsymbol{\lambda}, q, s) + \left( r_{\sigma_{n-1}} - q(1-s) \tilde{R}_{\boldsymbol{\sigma}^*}(n; \boldsymbol{\lambda}, q, s) \right) \lambda_{\sigma_{n-1}}$$

$$\leq q \tilde{R}_{\boldsymbol{\sigma}^*}(n; \boldsymbol{\lambda}, q, s) + \left( r_{\sigma_{n-1}} - q(1-s) \tilde{R}_{\boldsymbol{\sigma}^*}(n; \boldsymbol{\lambda}, q, s) \right) \tilde{\lambda}_{\sigma_{n-1}}$$

$$= r_{\sigma_{n-1}} \tilde{\lambda}_{\sigma_{n-1}} + \left( q - q(1-s) \tilde{\lambda}_{\sigma_{n-1}} \right) \tilde{R}_{\boldsymbol{\sigma}^*}(n; \boldsymbol{\lambda}, q, s) \tag{49}$$

$$= r_{\sigma_{n-1}} \tilde{\lambda}_{\sigma_{n-1}} + \left( q \left( 1 - \tilde{\lambda}_{\sigma_{n-1}} \right) + qs \tilde{\lambda}_{\sigma_{n-1}} \right) \tilde{R}_{\boldsymbol{\sigma}^*}(n; \boldsymbol{\lambda}, q, s)$$

$$\leq r_{\sigma_{n-1}} \tilde{\lambda}_{\sigma_{n-1}} + \left( \tilde{q} \left( 1 - \tilde{\lambda}_{\sigma_{n-1}} \right) + \tilde{q} \tilde{s} \tilde{\lambda}_{\sigma_{n-1}} \right) \tilde{R}_{\boldsymbol{\sigma}^*}(n; \boldsymbol{\lambda}, q, s)$$

$$\leq r_{\sigma_{n-1}} \tilde{\lambda}_{\sigma_{n-1}} + \left( \tilde{q} - \tilde{q}(1-\tilde{s}) \tilde{\lambda}_{\sigma_{n-1}} \right) \tilde{R}_{\boldsymbol{\sigma}^*}(n; \tilde{\boldsymbol{\lambda}}, \tilde{q}, \tilde{s}) = \tilde{R}_{\boldsymbol{\sigma}^*}(n-1; \tilde{\boldsymbol{\lambda}}, \tilde{q}, \tilde{s}).$$

As a result,

$$\text{Revenue}\left(\boldsymbol{\sigma}^*; \boldsymbol{\lambda}, q, s\right) = \tilde{R}_{\boldsymbol{\sigma}^*}(1; \boldsymbol{\lambda}, q, s) \leq \tilde{R}_{\boldsymbol{\sigma}^*}(1; \tilde{\boldsymbol{\lambda}}, \tilde{q}, \tilde{s}) = \text{Revenue}\left(\boldsymbol{\sigma}^*; \tilde{\boldsymbol{\lambda}}, \tilde{q}, \tilde{s}\right). \quad (50)$$

□

**Proposition C.2.** *Under Assumption 3.1, suppose* $\tilde{\boldsymbol{\lambda}} \succeq \boldsymbol{\lambda}$, $1 - \epsilon_Q \geq \tilde{q} \geq q$, *and* $\tilde{s}\tilde{q} \geq sq$, *then*

$$\text{Revenue}\left(\boldsymbol{\sigma}_t; \tilde{\boldsymbol{\lambda}}, \tilde{q}, \tilde{s}\right) - \text{Revenue}\left(\boldsymbol{\sigma}_t; \boldsymbol{\lambda}, q, s\right)$$

$$\leq \frac{r_{\max}}{\epsilon_Q} \sum_{k=1}^{N} \left( |q - \tilde{q}|\, \mathbb{E}\left[O_{t,k}^Q\right] + 3\left|\lambda_{\sigma_{t,k}} - \tilde{\lambda}_{\sigma_{t,k}}\right| \mathbb{E}\left[O_{t,k}^\Lambda\right] + |qs - \tilde{q}\tilde{s}|\, \mathbb{E}\left[O_{t,k}^W\right] \right). \quad (51)$$

*Proof of Proposition C.2.* For simplicity, we use $\boldsymbol{\sigma}$ to denote $\boldsymbol{\sigma}_t$ in this proof. According to Equation (30),

$$\text{Revenue}\left(\boldsymbol{\sigma}; \boldsymbol{\lambda}, q, s\right) = \sum_{k=1}^{N} r_{\sigma_k} \lambda_{\sigma_k} q^{k-1} \left( \sum_{u=0}^{k-1} s^u H_{\boldsymbol{\sigma}}(k, u; \boldsymbol{\lambda}) \right), \quad (52)$$

and $H_{\boldsymbol{\sigma}}(k, u; \boldsymbol{\lambda})$ is defined in Equation (29). As a result,

$$\text{Revenue}\left(\boldsymbol{\sigma}; \tilde{\boldsymbol{\lambda}}, \tilde{q}, \tilde{s}\right) - \text{Revenue}\left(\boldsymbol{\sigma}; \boldsymbol{\lambda}, q, s\right)$$

$$= \sum_{k=1}^{N} r_{\sigma_k} \tilde{\lambda}_{\sigma_k} \tilde{q}^{k-1} \left( \sum_{u=0}^{k-1} \tilde{s}^u H_{\boldsymbol{\sigma}}(k, u; \tilde{\boldsymbol{\lambda}}) \right) - \sum_{k=1}^{N} r_{\sigma_k} \lambda_{\sigma_k} q^{k-1} \left( \sum_{u=0}^{k-1} s^u H_{\boldsymbol{\sigma}}(k, u; \boldsymbol{\lambda}) \right) \quad (53)$$

$$= \sum_{k=1}^{N} r_{\sigma_k} \left( \tilde{\lambda}_{\sigma_k} \tilde{q}^{k-1} \left( \sum_{u=0}^{k-1} \tilde{s}^u H_{\boldsymbol{\sigma}}(k, u; \tilde{\boldsymbol{\lambda}}) \right) - \lambda_{\sigma_k} q^{k-1} \left( \sum_{u=0}^{k-1} s^u H_{\boldsymbol{\sigma}}(k, u; \boldsymbol{\lambda}) \right) \right).$$

Note that

$$\mathbb{E}\left[O_{t,k}^\Lambda\right] = P(V_t \geq k) \sum_{u=0}^{k-1} P(B_t \geq u + 1) H_{\boldsymbol{\sigma}}(k, u; \boldsymbol{\lambda}) = q^{k-1} \sum_{u=0}^{k-1} s^u H_{\boldsymbol{\sigma}}(k, u; \boldsymbol{\lambda}). \quad (54)$$

$$\mathbb{E}\left[O_{t,k}^Q\right] = \mathbb{E}\left[O_{t,k}^\Lambda\right] P\left(\gamma_{t,k} = 0\right) = (1 - \lambda_{\sigma_k}) q^{k-1} \sum_{u=0}^{k-1} s^u H_{\boldsymbol{\sigma}}(k, u; \boldsymbol{\lambda}). \quad (55)$$

$$\mathbb{E}\left[O_{t,k}^W\right] = \mathbb{E}\left[O_{t,k}^\Lambda\right] P\left(\gamma_{t,k} = 1\right) = \lambda_{\sigma_k} q^{k-1} \sum_{u=0}^{k-1} s^u H_{\boldsymbol{\sigma}}(k, u; \boldsymbol{\lambda}). \quad (56)$$

For any $1 \leq k \leq N$,

$$\left| \tilde{\lambda}_{\sigma_k} \tilde{q}^{k-1} \left( \sum_{u=0}^{k-1} \tilde{s}^u H_{\boldsymbol{\sigma}}(k, u; \tilde{\boldsymbol{\lambda}}) \right) - \lambda_{\sigma_k} q^{k-1} \left( \sum_{u=0}^{k-1} s^u H_{\boldsymbol{\sigma}}(k, u; \boldsymbol{\lambda}) \right) \right|$$

$$\leq \left| \tilde{\lambda}_{\sigma_k} - \lambda_{\sigma_k} \right| \mathbb{E}\left[O_{t,k}^\Lambda\right] + \tilde{\lambda}_{\sigma_k} \left| \tilde{q}^{k-1} \left( \sum_{u=0}^{k-1} \tilde{s}^u H_{\boldsymbol{\sigma}}(k, u; \tilde{\boldsymbol{\lambda}}) \right) - q^{k-1} \left( \sum_{u=0}^{k-1} s^u H_{\boldsymbol{\sigma}}(k, u; \boldsymbol{\lambda}) \right) \right| \quad (57)$$

$$\leq \left| \tilde{\lambda}_{\sigma_k} - \lambda_{\sigma_k} \right| \mathbb{E}\left[O_{t,k}^\Lambda\right] + \left| \tilde{q}^{k-1} \left( \sum_{u=0}^{k-1} \tilde{s}^u H_{\boldsymbol{\sigma}}(k, u; \tilde{\boldsymbol{\lambda}}) \right) - q^{k-1} \left( \sum_{u=0}^{k-1} s^u H_{\boldsymbol{\sigma}}(k, u; \boldsymbol{\lambda}) \right) \right|.$$

Here the first inequality is due to the triangle inequality of $|\cdot|$ and the second inequality is according to the fact that $\tilde{\lambda}_{\sigma_k} \leq 1$. Now for any $1 \leq k \leq N$, define

$$A_{\boldsymbol{\sigma}}(k; q, s, \boldsymbol{\lambda}) \triangleq q^{k-1} \left( \sum_{u=0}^{k-1} s^u H_{\boldsymbol{\sigma}}(k, u; \boldsymbol{\lambda}) \right). \quad (58)$$

We can get

$$
\begin{aligned}
&A_{\boldsymbol{\sigma}}(k; q, s, \boldsymbol{\lambda}) \\
&= q^{k-1} \left( \sum_{u=0}^{k-1} s^u H_{\boldsymbol{\sigma}}(k, u; \boldsymbol{\lambda}) \right) \\
&= q^{k-1} \left( \sum_{u=0}^{k-2} s^u \left(1 - \lambda_{\sigma_{k-1}}\right) H_{\boldsymbol{\sigma}}(k-1, u; \boldsymbol{\lambda}) + \sum_{u=1}^{k-1} s^u \lambda_{\sigma_{k-1}} H_{\boldsymbol{\sigma}}(k-1, u-1; \boldsymbol{\lambda}) \right) \quad (59) \\
&= q^{k-1} \left( \sum_{u=0}^{k-2} s^u \left(1 - \lambda_{\sigma_{k-1}}\right) H_{\boldsymbol{\sigma}}(k-1, u; \boldsymbol{\lambda}) + s \sum_{u=0}^{k-2} s^u \lambda_{\sigma_{k-1}} H_{\boldsymbol{\sigma}}(k-1, u; \boldsymbol{\lambda}) \right) \\
&= q \left(1 - \lambda_{\sigma_{k-1}} + s\lambda_{\sigma_{k-1}}\right) A_{\boldsymbol{\sigma}}(k-1; q, s, \boldsymbol{\lambda}).
\end{aligned}
$$

As a result, for any $1 < k \le N$,

$$
\begin{aligned}
&\left| A_{\boldsymbol{\sigma}}(k; q, s, \boldsymbol{\lambda}) - A_{\boldsymbol{\sigma}}(k; \tilde{q}, \tilde{s}, \tilde{\boldsymbol{\lambda}}) \right| \\
&= \left| q \left(1 - \lambda_{\sigma_{k-1}} + s\lambda_{\sigma_{k-1}}\right) A_{\boldsymbol{\sigma}}(k-1; q, s, \boldsymbol{\lambda}) - \tilde{q} \left(1 - \tilde{\lambda}_{\sigma_{k-1}} + \tilde{s}\tilde{\lambda}_{\sigma_{k-1}}\right) A_{\boldsymbol{\sigma}}(k-1; \tilde{q}, \tilde{s}, \tilde{\boldsymbol{\lambda}}) \right| \\
&\le \left| q \left(1 - \lambda_{\sigma_{k-1}} + s\lambda_{\sigma_{k-1}}\right) - \tilde{q} \left(1 - \tilde{\lambda}_{\sigma_{k-1}} + \tilde{s}\tilde{\lambda}_{\sigma_{k-1}}\right) \right| A_{\boldsymbol{\sigma}}(k-1; q, s, \boldsymbol{\lambda}) \\
&\quad + \tilde{q} \left(1 - \tilde{\lambda}_{\sigma_{k-1}} + \tilde{s}\tilde{\lambda}_{\sigma_{k-1}}\right) \left| A_{\boldsymbol{\sigma}}(k-1; q, s, \boldsymbol{\lambda}) - A_{\boldsymbol{\sigma}}(k-1; \tilde{q}, \tilde{s}, \tilde{\boldsymbol{\lambda}}) \right|.
\end{aligned} \quad (60)
$$

Here the last inequality is due the triangle inequality of $|\cdot|$. Note that

$$
\begin{aligned}
&\left| q \left(1 - \lambda_{\sigma_{k-1}} + s\lambda_{\sigma_{k-1}}\right) - \tilde{q} \left(1 - \tilde{\lambda}_{\sigma_{k-1}} + \tilde{s}\tilde{\lambda}_{\sigma_{k-1}}\right) \right| A_{\boldsymbol{\sigma}}(k-1; q, s, \boldsymbol{\lambda}) \\
&\le \left( \left| q \left(1 - \lambda_{\sigma_{k-1}}\right) - \tilde{q} \left(1 - \tilde{\lambda}_{\sigma_{k-1}}\right) \right| + \left| qs\lambda_{\sigma_{k-1}} - \tilde{q}\tilde{s}\tilde{\lambda}_{\sigma_{k-1}} \right| \right) A_{\boldsymbol{\sigma}}(k-1; q, s, \boldsymbol{\lambda}) \\
&\le \left( |q - \tilde{q}| \left(1 - \lambda_{\sigma_{k-1}}\right) + \tilde{q} \left| \lambda_{\sigma_{k-1}} - \tilde{\lambda}_{\sigma_{k-1}} \right| + |qs - \tilde{q}\tilde{s}| \lambda_{\sigma_{k-1}} + \tilde{q}\tilde{s} \left| \lambda_{\sigma_{k-1}} - \tilde{\lambda}_{\sigma_{k-1}} \right| \right) A_{\boldsymbol{\sigma}}(k-1; q, s, \boldsymbol{\lambda}) \\
&\le \left( |q - \tilde{q}| \left(1 - \lambda_{\sigma_{k-1}}\right) + 2 \left| \lambda_{\sigma_{k-1}} - \tilde{\lambda}_{\sigma_{k-1}} \right| + |qs - \tilde{q}\tilde{s}| \lambda_{\sigma_{k-1}} \right) A_{\boldsymbol{\sigma}}(k-1; q, s, \boldsymbol{\lambda}) \\
&= |q - \tilde{q}| \, \mathbb{E}\left[ O_{t,k-1}^Q \right] + 2 \left| \lambda_{\sigma_{k-1}} - \tilde{\lambda}_{\sigma_{k-1}} \right| \mathbb{E}\left[ O_{t,k-1}^{\Lambda} \right] + |qs - \tilde{q}\tilde{s}| \, \mathbb{E}\left[ O_{t,k-1}^W \right].
\end{aligned}
$$
$$(61)$$

Here the first two inequalities are due to the triangle inequality of $|\cdot|$ and the third inequality is due to that fact that $\tilde{q}, \tilde{s} \le 1$. Combining Equations (60) and (61), we can get

$$
\begin{aligned}
&\left| A_{\boldsymbol{\sigma}}(k; q, s, \boldsymbol{\lambda}) - A_{\boldsymbol{\sigma}}(k; \tilde{q}, \tilde{s}, \tilde{\boldsymbol{\lambda}}) \right| \\
&\le \sum_{u=1}^{k-1} \left( \left( |q - \tilde{q}| \, \mathbb{E}\left[ O_{t,u}^Q \right] + 2 \left| \lambda_{\sigma_u} - \tilde{\lambda}_{\sigma_u} \right| \mathbb{E}\left[ O_{t,u}^{\Lambda} \right] + |qs - \tilde{q}\tilde{s}| \, \mathbb{E}\left[ O_{t,u}^W \right] \right) \left( \prod_{i=u+1}^{k-1} \tilde{q} \left(1 - \tilde{\lambda}_{\sigma_i} + \tilde{s}\tilde{\lambda}_{\sigma_i}\right) \right) \right) \\
&\le \sum_{u=1}^{k-1} \left( \left( |q - \tilde{q}| \, \mathbb{E}\left[ O_{t,u}^Q \right] + 2 \left| \lambda_{\sigma_u} - \tilde{\lambda}_{\sigma_u} \right| \mathbb{E}\left[ O_{t,u}^{\Lambda} \right] + |qs - \tilde{q}\tilde{s}| \, \mathbb{E}\left[ O_{t,u}^W \right] \right) \left( \prod_{i=u+1}^{k-1} \tilde{q} \right) \right) \\
&= \sum_{u=1}^{k-1} \left( |q - \tilde{q}| \, \mathbb{E}\left[ O_{t,u}^Q \right] + 2 \left| \lambda_{\sigma_u} - \tilde{\lambda}_{\sigma_u} \right| \mathbb{E}\left[ O_{t,u}^{\Lambda} \right] + |qs - \tilde{q}\tilde{s}| \, \mathbb{E}\left[ O_{t,u}^W \right] \right) \tilde{q}^{k-u-1}.
\end{aligned}
$$
$$(62)$$

Combining the above inequality with Equations (53) and (57), we can get that

$$
\text{Revenue}\left(\boldsymbol{\sigma}; \tilde{\boldsymbol{\lambda}}, \tilde{q}, \tilde{s}\right) - \text{Revenue}\left(\boldsymbol{\sigma}; \boldsymbol{\lambda}, q, s\right)
$$

$$
\leq \sum_{k=1}^{N} r_{\max} \left|\tilde{\lambda}_{\sigma_k} - \lambda_{\sigma_k}\right| \mathbb{E}\left[O_{t,k}^{\Lambda}\right]
$$

$$
+ \sum_{k=2}^{N} r_{\sigma_k} \tilde{\lambda}_{\sigma_k} \left(\sum_{u=1}^{k-1} \left(|q - \tilde{q}|\, \mathbb{E}\left[O_{t,u}^{Q}\right] + 2\left|\lambda_{\sigma_u} - \tilde{\lambda}_{\sigma_u}\right| \mathbb{E}\left[O_{t,u}^{\Lambda}\right] + |qs - \tilde{q}\tilde{s}|\, \mathbb{E}\left[O_{t,u}^{W}\right]\right) \tilde{q}^{k-u-1}\right)
$$

$$
= \sum_{k=1}^{N} r_{\max} \left|\tilde{\lambda}_{\sigma_k} - \lambda_{\sigma_k}\right| \mathbb{E}\left[O_{t,k}^{\Lambda}\right]
$$

$$
+ \sum_{k=1}^{N-1} \left(|q - \tilde{q}|\, \mathbb{E}\left[O_{t,k}^{Q}\right] + 2\left|\lambda_{\sigma_k} - \tilde{\lambda}_{\sigma_k}\right| \mathbb{E}\left[O_{t,k}^{\Lambda}\right] + |qs - \tilde{q}\tilde{s}|\, \mathbb{E}\left[O_{t,k}^{W}\right]\right) \left(\sum_{i=k+1}^{N} r_{\sigma_i} \tilde{\lambda}_{\sigma_i} \tilde{q}^{i-k-1}\right)
$$

$$
\leq \sum_{k=1}^{N} r_{\max} \left|\tilde{\lambda}_{\sigma_k} - \lambda_{\sigma_k}\right| \mathbb{E}\left[O_{t,k}^{\Lambda}\right]
$$

$$
+ \sum_{k=1}^{N-1} r_{\max} \left(|q - \tilde{q}|\, \mathbb{E}\left[O_{t,k}^{Q}\right] + 2\left|\lambda_{\sigma_k} - \tilde{\lambda}_{\sigma_k}\right| \mathbb{E}\left[O_{t,k}^{\Lambda}\right] + |qs - \tilde{q}\tilde{s}|\, \mathbb{E}\left[O_{t,k}^{W}\right]\right) \left(\sum_{i=k+1}^{N} \tilde{q}^{i-k-1}\right)
$$

$$
\leq \sum_{k=1}^{N} r_{\max} \left|\tilde{\lambda}_{\sigma_k} - \lambda_{\sigma_k}\right| \mathbb{E}\left[O_{t,k}^{\Lambda}\right]
$$

$$
+ \sum_{k=1}^{N-1} r_{\max} \left(|q - \tilde{q}|\, \mathbb{E}\left[O_{t,k}^{Q}\right] + 2\left|\lambda_{\sigma_k} - \tilde{\lambda}_{\sigma_k}\right| \mathbb{E}\left[O_{t,k}^{\Lambda}\right] + |qs - \tilde{q}\tilde{s}|\, \mathbb{E}\left[O_{t,k}^{W}\right]\right) \frac{1}{\epsilon_Q}
$$

$$
\leq \frac{r_{\max}}{\epsilon_Q} \sum_{k=1}^{N} \left(|q - \tilde{q}|\, \mathbb{E}\left[O_{t,k}^{Q}\right] + 3\left|\lambda_{\sigma_k} - \tilde{\lambda}_{\sigma_k}\right| \mathbb{E}\left[O_{t,k}^{\Lambda}\right] + |qs - \tilde{q}\tilde{s}|\, \mathbb{E}\left[O_{t,k}^{W}\right]\right).
$$

$$(63)$$

Now the claim follows. $\qquad\square$

*Proof of Theorem 5.3.* For each timestamp $t$, define the favorable event as

$$
\xi_t \triangleq \left\{ |q - \hat{q}_t| \leq \sqrt{\frac{2\log t}{D_t^Q}}, |qs - \hat{w}_t| \leq \sqrt{\frac{2\log t}{D_t^W}}, \left|\lambda_k - \hat{\lambda}_{t,k}\right| \leq \sqrt{\frac{2\log t}{C_{t,k}}}, \forall k \in [N] \right\}. \quad (64)
$$

Let $\bar{\xi}_t$ denote the complement event of $\xi_t$.

$$
\mathbb{E}[\text{Reg}_T] = \sum_{t=1}^{T} \mathbb{E}\left[\left(\text{Revenue}\left(\boldsymbol{\sigma}^*; \boldsymbol{\lambda}, q, s\right) - \text{Revenue}\left(\boldsymbol{\sigma}_t; \boldsymbol{\lambda}, q, s\right)\right) \mathbb{I}[\xi_{t-1}]\right]
$$

$$
+ \sum_{t=1}^{T} \mathbb{E}\left[\left(\text{Revenue}\left(\boldsymbol{\sigma}^*; \boldsymbol{\lambda}, q, s\right) - \text{Revenue}\left(\boldsymbol{\sigma}_t; \boldsymbol{\lambda}, q, s\right)\right) \mathbb{I}\left[\bar{\xi}_{t-1}\right]\right].
$$

$$(65)$$

On the one hand, consider the first term in the RHS of Equation (65), when the event $\xi_{t-1}$ is satisfied, we have $q \leq \tilde{q}_{t-1}$, $qs \leq \tilde{w}_{t-1}$, and $\lambda_k \leq \tilde{\lambda}_{t,k}, \forall k \in [N]$. As a result, according to Proposition C.1,

$$
\text{Revenue}\left(\boldsymbol{\sigma}^*; \boldsymbol{\lambda}, q, s\right) \leq \text{Revenue}\left(\boldsymbol{\sigma}^*; \tilde{\boldsymbol{\lambda}}_{t-1}, \tilde{q}_{t-1}, \tilde{s}_{t-1}\right) \leq \text{Revenue}\left(\boldsymbol{\sigma}_t; \tilde{\boldsymbol{\lambda}}_{t-1}, \tilde{q}_{t-1}, \tilde{s}_{t-1}\right). \quad (66)
$$

Hence, according to Proposition C.2,

$$\sum_{t=1}^{T} \mathbb{E}\left[\left(\text{Revenue}\left(\boldsymbol{\sigma}^*; \boldsymbol{\lambda}, q, s\right) - \text{Revenue}\left(\boldsymbol{\sigma}_t; \boldsymbol{\lambda}, q, s\right)\right) \mathbb{I}[\xi_{t-1}]\right]$$

$$\leq \sum_{t=1}^{T} \mathbb{E}\left[\left(\text{Revenue}\left(\boldsymbol{\sigma}_t; \tilde{\boldsymbol{\lambda}}_{t-1}, \tilde{q}_{t-1}, \tilde{s}_{t-1}\right) - \text{Revenue}\left(\boldsymbol{\sigma}_t; \boldsymbol{\lambda}, q, s\right)\right) \mathbb{I}[\xi_{t-1}]\right]$$

$$\leq \frac{r_{\max}}{\epsilon_Q} \sum_{t=1}^{T} \mathbb{E}\left[\sum_{k=1}^{N} \left(|q - \tilde{q}_{t-1}| \mathbb{E}\left[O_{t,k}^Q\right] + 3\left|\lambda_{\sigma_{t,k}} - \tilde{\lambda}_{t-1,\sigma_{t,k}}\right| \mathbb{E}\left[O_{t,k}^\Lambda\right] + |qs - \tilde{q}_{t-1}\tilde{s}_{t-1}| \mathbb{E}\left[O_{t,k}^W\right]\right) \mathbb{I}[\xi_{t-1}]\right]$$

$$\leq \frac{2r_{\max}}{\epsilon_Q} \sum_{t=1}^{T} \mathbb{E}\left[\sum_{k=1}^{N} \left(\min\left\{\sqrt{\frac{2\log t}{D_{t-1}^Q}}, 1\right\} O_{t,k}^Q + 3\min\left\{\sqrt{\frac{2\log t}{C_{t-1,\sigma_{t,k}}}}, 1\right\} O_{t,k}^\Lambda + \min\left\{\sqrt{\frac{2\log t}{D_{t-1}^W}}, 1\right\} O_{t,k}^W\right)\right]$$

$$\leq \frac{2r_{\max}\sqrt{2\log T}}{\epsilon_Q} \sum_{t=1}^{T} \mathbb{E}\left[\sum_{k=1}^{N} \left(\min\left\{\sqrt{\frac{1}{D_{t-1}^Q}}, 1\right\} O_{t,k}^Q + 3\min\left\{\sqrt{\frac{1}{C_{t-1,\sigma_{t,k}}}}, 1\right\} O_{t,k}^\Lambda + \min\left\{\sqrt{\frac{1}{D_{t-1}^W}}, 1\right\} O_{t,k}^W\right)\right].$$

$$(67)$$

Let $\kappa_t^Q \triangleq \sum_{k=1}^{N} O_{t,k}^Q \leq N$ and we have $D_t^Q = D_{t-1}^Q + \kappa_t^Q$. As a result,

$$\mathbb{E}\left[\sum_{t=1}^{T}\sum_{k=1}^{N} \min\left\{\sqrt{\frac{1}{D_{t-1}^Q}}, 1\right\} O_{t,k}^Q\right]$$

$$= \mathbb{E}\left[\sum_{t=1}^{T} \min\left\{\sqrt{\frac{1}{D_{t-1}^Q}}, 1\right\} \kappa_t^Q\right]$$

$$\leq 1 + N + \sum_{t=2}^{T} N\sqrt{\frac{1}{(t-1)N}}$$

$$\leq 1 + N + \sqrt{N}\left(1 + \int_{t=1}^{T-1} \sqrt{\frac{1}{t}}\mathrm{d}t\right) \leq C_1 N + C_2\sqrt{TN},$$

$$(68)$$

for some constants $C_1$ and $C_2$. Similarly, we have

$$\mathbb{E}\left[\sum_{t=1}^{T}\sum_{k=1}^{N} \min\left\{\sqrt{\frac{1}{D_{t-1}^W}}, 1\right\} O_{t,k}^W\right] \leq C_1 N + C_2\sqrt{TN}. \qquad (69)$$

In addition, let $o_{t,k}^\Lambda \triangleq O_{t,\sigma_{t,k}^{-1}}^\Lambda$ represent whether the product $k$ is viewed in the $t$-th timestamp. As a result, $C_{t,k} = C_{t-1,k} + o_{t,k}^\Lambda$. Therefore,

$$\mathbb{E}\left[\sum_{t=1}^{T}\sum_{k=1}^{N} \min\left\{\sqrt{\frac{1}{C_{t-1,\sigma_{t,k}}}}, 1\right\} O_{t,k}^\Lambda\right]$$

$$= \mathbb{E}\left[\sum_{k=1}^{N}\sum_{t=1}^{T} \min\left\{\sqrt{\frac{1}{C_{t-1,k}}}, 1\right\} o_{t,k}^\Lambda\right]$$

$$\leq \mathbb{E}\left[\sum_{k=1}^{N}\left(1 + \sum_{t=1}^{T} \sqrt{\frac{1}{t}}\right)\right] \leq C_3 N\sqrt{T},$$

$$(70)$$

for a constant $C_3$. On the other hand, according to Proposition 5.2,

$$\sum_{t=1}^{T} \mathbb{E}\left[\left(\text{Revenue}\left(\boldsymbol{\sigma}^*; \boldsymbol{\lambda}, q, s\right) - \text{Revenue}\left(\boldsymbol{\sigma}_t; \boldsymbol{\lambda}, q, s\right)\right) \mathbb{I}\left[\bar{\xi}_{t-1}\right]\right]$$

$$\leq \text{Revenue}\left(\boldsymbol{\sigma}^*; \boldsymbol{\lambda}, q, s\right) \sum_{t=1}^{T} \mathbb{E}\left[\mathbb{I}\left[\bar{\xi}_{t-1}\right]\right] \leq \frac{r_{\max}}{\epsilon_Q} \sum_{t=1}^{T} \mathbb{E}\left[\mathbb{I}\left[\bar{\xi}_{t-1}\right]\right] \qquad (71)$$

$$\leq \frac{r_{\max}}{\epsilon_Q}\left(1 + \sum_{t=1}^{T} \frac{6N}{t^3}\right) \leq C_4 \cdot \frac{Nr_{\max}}{\epsilon_Q}.$$

for a constant $C_4$. Now the claim follows after combining Equations (67), (68), (69), (70), and (71). $\qquad\square$

## C.5    Proof of Proposition 5.4

*Proof.* The claim follows from standard routines in linear bandit methods. We first consider the $\hat{\beta}_t^\Lambda - \beta^\Lambda$ term. We can view the tuple $(t,k)$ as the new timestamp. Let $\epsilon_{t,k} \triangleq y_{t,\sigma_{t,k}}^\top \beta^\Lambda - \gamma_{t,k}$. According to Lemma 5.1, $\mathbb{E}[\epsilon_{t,k}|y_{t,\sigma_{t,k}}, \mathcal{H}_{t,k}] = 0$ where $\mathcal{H}_{t,k}$ denotes the historical observations before the $t$-th consumer views the $k$-th product. In addition, it is bounded in $[q_t - 1, q_t]$ and hence sub-gaussian with $R = 1/2$. As a result, according to [1, Theorem 2], under Assumption 5.2 with parameter $U$, for any $\delta \in (0,1)$, with probability at least $1 - \delta$, for any $t \geq 0$,

$$
\begin{aligned}
\left\| \hat{\beta}_t^\Lambda - \beta^\Lambda \right\|_{\Sigma_t^\Lambda} &\leq \frac{1}{2} \sqrt{2 \log \frac{\det \left( \Sigma_t^\Lambda \right)^{1/2} \det \left( \alpha_\Lambda \mathbf{I}_{m_y} \right)^{-1/2}}{\delta}} + \alpha_\Lambda^{1/2} U \\
&= \frac{1}{2} \sqrt{\log \frac{\det \left( \Sigma_t^\Lambda \right)}{\delta^2 \alpha_\Lambda^{m_y}}} + \alpha_\Lambda^{1/2} U.
\end{aligned}
\tag{72}
$$

According to [1, Lemma 10], we have

$$
\det \left( \Sigma_t^\Lambda \right) \leq \left( \alpha_\Lambda + \frac{Nt}{m_y} \right)^{m_y}.
\tag{73}
$$

As a result, by letting $\delta = t^{-2}$, for all $t > 0$, with probability at least $1 - t^{-2}$,

$$
\left\| \hat{\beta}_t^\Lambda - \beta^\Lambda \right\|_{\Sigma_t^\Lambda} \leq \frac{1}{2} \sqrt{m_y \log \left( 1 + \frac{Nt}{m_y \alpha_\Lambda} \right) + 4 \log t} + \alpha_\Lambda^{1/2} U \leq \tau(t, m_y, \alpha_\Lambda).
\tag{74}
$$

Similarly, we have that with probability at least $1 - t^{-2}$,

$$
\left\| \hat{\beta}_t^Q - \beta^Q \right\|_{\Sigma_t^Q} \leq \frac{1}{2} \sqrt{m_x \log \left( 1 + \frac{Nt}{m_x \alpha_Q} \right) + 4 \log t} + \alpha_Q^{1/2} U \leq \tau(t, m_x, \alpha_Q).
\tag{75}
$$

In addition, because

$$
\|x_t\| \leq 1 \implies \|z_t\| \leq 1 \quad \text{and} \quad \|\beta^S\|, \|\beta^Q\| \leq U \implies \|\beta^W\| \leq U^2,
\tag{76}
$$

we have that with probability at least $1 - t^{-2}$,

$$
\left\| \hat{\beta}_t^W - \beta^W \right\|_{\Sigma_t^W} \leq \frac{1}{2} \sqrt{m_x^2 \log \left( 1 + \frac{Nt}{m_x^2 \alpha_W} \right) + 4 \log t} + \alpha_W^{1/2} U^2 \leq \tau(t, m_x^2, \alpha_W).
\tag{77}
$$

Now the claim follows. $\qquad\square$

## C.6    Proof of Theorem 5.5

*Proof.* Similar to the proof of Theorem 5.3, for each timestamp $t$, define the favorable event as

$$
\xi_t \triangleq \left\{ \left\| \hat{\beta}_t^\Lambda - \beta^\Lambda \right\|_{\Sigma_t^\Lambda} \leq \tau(t, m_y, \alpha_\Lambda), \left\| \hat{\beta}_t^Q - \beta^Q \right\|_{\Sigma_t^Q} \leq \tau(t, m_x, \alpha_Q), \text{and} \left\| \hat{\beta}_t^W - \beta^W \right\|_{\Sigma_t^W} \leq \tau(t, m_x^2, \alpha_W) \right\}.
\tag{78}
$$

According to Proposition 5.4, we have $P(\xi_t) \geq 1 - 3t^{-2}$. Let $\bar{\xi}_t$ denote the complement event of $\xi_t$.

$$
\begin{aligned}
\mathbb{E}[\mathrm{Reg}_T | \boldsymbol{X}_T, \boldsymbol{Y}_T] = &\sum_{t=1}^T \mathbb{E} \left[ \left( \mathrm{Revenue}\left(\boldsymbol{\sigma}_t^*; \boldsymbol{\lambda}_t, q_t, s_t \right) - \mathrm{Revenue}\left(\boldsymbol{\sigma}_t; \boldsymbol{\lambda}_t, q_t, s_t \right) \right) \mathbb{I}[\xi_{t-1}] \right] \\
&+ \sum_{t=1}^T \mathbb{E} \left[ \left( \mathrm{Revenue}\left(\boldsymbol{\sigma}_t^*; \boldsymbol{\lambda}_t, q_t, s_t \right) - \mathrm{Revenue}\left(\boldsymbol{\sigma}_t; \boldsymbol{\lambda}_t, q_t, s_t \right) \right) \mathbb{I}\left[\bar{\xi}_{t-1}\right] \right].
\end{aligned}
\tag{79}
$$

On the one hand, consider the first term in the RHS of Equation (79), when the event $\xi_t$ is satisfied, we can show that $\tilde{q}_t \geq q_t$. On the one hand, if $\tilde{q}_t = 1 - \epsilon_Q$, it is obvious that $\tilde{q}_t \geq q_t$. On the other

hand, otherwise,

$$
\begin{aligned}
\tilde{q}_t - q_t &= x_t^\top \left( \hat{\beta}_{t-1}^Q - \beta^Q \right) + \tau(t-1, m_x, \alpha_Q) \|x_t\|_{(\Sigma_{t-1}^Q)^{-1}} \\
&= x_t^\top \left( \Sigma_{t-1}^Q \right)^{-1/2} \left( \Sigma_{t-1}^Q \right)^{1/2} \left( \hat{\beta}_{t-1}^Q - \beta^Q \right) + \tau(t-1, m_x, \alpha_Q) \|x_t\|_{(\Sigma_{t-1}^Q)^{-1}} \\
&= \left( \left( \Sigma_{t-1}^Q \right)^{-1/2} x_t \right)^\top \left( \Sigma_{t-1}^Q \right)^{1/2} \left( \hat{\beta}_{t-1}^Q - \beta^Q \right) + \tau(t-1, m_x, \alpha_Q) \|x_t\|_{(\Sigma_{t-1}^Q)^{-1}} \\
&\geq - \|x_t\|_{(\Sigma_{t-1}^Q)^{-1}} \left\| \hat{\beta}_{t-1}^Q - \beta^Q \right\|_{\Sigma_{t-1}^Q} + \tau(t-1, m_x, \alpha_Q) \|x_t\|_{(\Sigma_{t-1}^Q)^{-1}} \geq 0.
\end{aligned}
\tag{80}
$$

Similarly, we have that $w_t \leq \tilde{w}_t$ and $\lambda_{t,k} \leq \tilde{\lambda}_{t,k}, \forall k \in [N]$. As a result, according to Proposition C.1,

$$
\text{Revenue}\,(\boldsymbol{\sigma}_t^*; \boldsymbol{\lambda}_t, q_t, s_t) \leq \text{Revenue}\left(\boldsymbol{\sigma}_t^*; \tilde{\boldsymbol{\lambda}}_t, \tilde{q}_t, \tilde{s}_t\right) \leq \text{Revenue}\left(\boldsymbol{\sigma}_t; \tilde{\boldsymbol{\lambda}}_t, \tilde{q}_t, \tilde{s}_t\right). \tag{81}
$$

Hence, according to Proposition C.2,

$$
\begin{aligned}
& \sum_{t=1}^T \mathbb{E}\left[(\text{Revenue}\,(\boldsymbol{\sigma}_t^*; \boldsymbol{\lambda}_t, q_t, s_t) - \text{Revenue}\,(\boldsymbol{\sigma}_t; \boldsymbol{\lambda}_t, q_t, s_t))\,\mathbb{I}[\xi_t]\right] \\
& \leq \sum_{t=1}^T \mathbb{E}\left[\left(\text{Revenue}\left(\boldsymbol{\sigma}_t; \tilde{\boldsymbol{\lambda}}_t, \tilde{q}_t, \tilde{s}_t\right) - \text{Revenue}\,(\boldsymbol{\sigma}_t; \boldsymbol{\lambda}_t, q_t, s_t)\right)\mathbb{I}[\xi_t]\right] \\
& \leq \frac{r_{\max}}{\epsilon_Q} \sum_{t=1}^T \mathbb{E}\left[\sum_{k=1}^N \left(|q_t - \tilde{q}_t|\,\mathbb{E}\left[O_{t,k}^Q\right] + 3\left|\lambda_{t,\sigma_{t,k}} - \tilde{\lambda}_{t,\sigma_{t,k}}\right|\mathbb{E}\left[O_{t,k}^\Lambda\right] + |w_t - \tilde{w}_t|\,\mathbb{E}\left[O_{t,k}^W\right]\right)\mathbb{I}[\xi_t]\right] \\
& \leq \frac{2 r_{\max}}{\epsilon_Q} \sum_{t=1}^T \mathbb{E}\Bigg[\sum_{k=1}^N \Bigg(\tau(t-1, m_x, \alpha_Q)\,\|x_t\|_{(\Sigma_{t-1}^Q)^{-1}}\,O_{t,k}^Q + 3\tau(t-1, m_y, \alpha_\Lambda)\,\|y_{t,\sigma_{t,k}}\|_{(\Sigma_{t-1}^\Lambda)^{-1}}\,O_{t,k}^\Lambda \\
& \qquad\qquad\qquad + \tau(t-1, m_x^2, \alpha_W)\,\|z_t\|_{(\Sigma_{t-1}^W)^{-1}}\,O_{t,k}^W\Bigg)\mathbb{I}[\xi_{t-1}]\Bigg].
\end{aligned}
\tag{82}
$$

We consider the $\sum_{t=1}^T \mathbb{E}\left[\sum_{k=1}^N \tau(t-1, m_y, \alpha_\Lambda)\,\|y_{t,\sigma_{t,k}}\|_{(\Sigma_{t-1}^\Lambda)^{-1}}\,O_{t,k}^\Lambda\right]$ term first and this part is similar to the proof of [51, Lemma 1]. Define $g_{t,k} = \|y_{t,\sigma_{t,k}}\|_{(\Sigma_{t-1}^\Lambda)^{-1}}$ for all $t \in [T]$, $k \in [N]$ that satisfy $O_{t,k}^\Lambda = 1$. Because

$$
\Sigma_t^\Lambda = \sum_{u=1}^t \sum_{k=1}^{\Phi_u} y_{u,\sigma_{u,k}} y_{u,\sigma_{u,k}}^\top + \alpha_\Lambda \mathbf{I}_{m_y} = \Sigma_{t-1}^\Lambda + \sum_{k=1}^{\Phi_t} y_{t,\sigma_{t,k}} y_{t,\sigma_{t,k}}^\top, \quad \Sigma_0^\Lambda = \alpha_\Lambda \mathbf{I}_{m_y}, \tag{83}
$$

we have for all $1 \leq k \leq \Phi_t$,

$$
\det\left(\Sigma_t^\Lambda\right) \geq \det\left(\Sigma_{t-1}^\Lambda + y_{t,\sigma_{t,k}} y_{t,\sigma_{t,k}}^\top\right) = \det\left(\Sigma_{t-1}^\Lambda\right)\left(1 + g_{t,k}^2\right). \tag{84}
$$

As a result,

$$
\left(\frac{\det\left(\Sigma_t^\Lambda\right)}{\det\left(\Sigma_{t-1}^\Lambda\right)}\right)^N \geq \left(\frac{\det\left(\Sigma_t^\Lambda\right)}{\det\left(\Sigma_{t-1}^\Lambda\right)}\right)^{\Phi_t} = \prod_{k=1}^{\Phi_t}(1 + g_{t,k}^2). \tag{85}
$$

Hence,

$$
\left(\det\left(\Sigma_t^\Lambda\right)\right)^N \geq \alpha_\Lambda^{N m_y} \prod_{u=1}^t \prod_{k=1}^{\Phi_t}(1 + g_{u,k}^2). \tag{86}
$$

On the other hand, due to Assumption 5.2,

$$
\begin{aligned}
\text{trace}\left(\Sigma_t^\Lambda\right) &= \text{trace}\left(\sum_{u=1}^t \sum_{k=1}^{\Phi_u} y_{u,\sigma_{u,k}} y_{u,\sigma_{u,k}}^\top + \alpha_\Lambda \mathbf{I}_{m_y}\right) = m_y \alpha_\Lambda + \sum_{u=1}^t \sum_{k=1}^{\Phi_t} \|y_{u,\sigma_{u,k}}\|^2 \\
&\leq m_y \alpha_\Lambda + tN.
\end{aligned}
\tag{87}
$$

According to the [1, Lemma 10], we have $\text{trace}(\Sigma_t^\Lambda)/m_y \geq (\det(\Sigma_t^\Lambda))^{1/m_y}$. As a result,

$$\left(\frac{m_y\alpha_\Lambda + tN}{m_y}\right)^{Nm_y} \geq \alpha_\Lambda^{Nm_y} \prod_{u=1}^{t} \prod_{k=1}^{\Phi_t} (1 + g_{u,k}^2). \tag{88}$$

By taking the logarithm, we can get that

$$\sum_{u=1}^{t} \sum_{k=1}^{\Phi_t} \log(1 + g_{u,k}^2) \leq Nm_y \log\left(1 + \frac{tN}{m_y\alpha_\Lambda}\right). \tag{89}$$

Since

$$\begin{aligned}
g_{t,k}^2 &= \|y_{t,\sigma_{t,k}}\|_{(\Sigma_{t-1}^\Lambda)^{-1}}^2 = y_{t,\sigma_{t,k}}^\top \left(\Sigma_{t-1}^\Lambda\right)^{-1} y_{t,\sigma_{t,k}} \\
&\leq y_{t,\sigma_{t,k}}^\top \left(\Sigma_0^\Lambda\right)^{-1} y_{t,\sigma_{t,k}} = \alpha_\Lambda^{-1} \|y_{t,\sigma_{t,k}}\|^2 \leq \alpha_\Lambda^{-1},
\end{aligned} \tag{90}$$

we have $g_{t,k}^2 \leq \alpha_\Lambda^{-1} \log(1 + g_{t,k}^2)/\log(1 + \alpha_\Lambda^{-1})$. As a result,

$$\sum_{u=1}^{t} \sum_{k=1}^{\Phi_t} g_{u,k}^2 \leq \frac{1}{\alpha_\Lambda \log(1 + \alpha_\Lambda^{-1})} \sum_{u=1}^{t} \sum_{k=1}^{\Phi_t} \log(1 + g_{u,k}^2) \leq \frac{Nm_y \log\left(1 + \frac{tN}{m_y\alpha_\Lambda}\right)}{\alpha_\Lambda \log(1 + \alpha_\Lambda^{-1})}. \tag{91}$$

Hence, according to the Cauchy–Schwarz inequality, we have

$$\begin{aligned}
\sum_{t=1}^{T} \sum_{k=1}^{N} \|y_{t,\sigma_{t,k}}\|_{(\Sigma_{t-1}^\Lambda)^{-1}} O_{t,k}^\Lambda &\leq \sqrt{\left(\sum_{t=1}^{T} \sum_{k=1}^{\Phi_t} g_{t,k}^2\right)\left(\sum_{t=1}^{T} \sum_{k=1}^{\Phi_t} O_{t,k}^\Lambda\right)} \\
&\leq \sqrt{\frac{TN^2 m_y \log\left(1 + \frac{TN}{m_y\alpha_\Lambda}\right)}{\alpha_\Lambda \log(1 + \alpha_\Lambda^{-1})}}.
\end{aligned} \tag{92}$$

As a result, we have

$$\begin{aligned}
&\sum_{t=1}^{T} \sum_{k=1}^{N} \tau(t-1, m_y, \alpha_\Lambda) \|y_{t,k}\|_{(\Sigma_{t-1}^\Lambda)^{-1}} O_{t,k}^\Lambda \\
&\leq \tau(T-1, m_y, \alpha_\Lambda) \sum_{t=1}^{T} \sum_{k=1}^{N} \|y_{t,k}\|_{(\Sigma_{t-1}^\Lambda)^{-1}} O_{t,k}^\Lambda \\
&\leq \tau(T-1, m_y, \alpha_\Lambda) \sqrt{\frac{TN^2 m_y \log\left(1 + \frac{TN}{m_y\alpha_\Lambda}\right)}{\alpha_\Lambda \log(1 + \alpha_\Lambda^{-1})}} \leq \chi(T, m_y, \alpha_\Lambda).
\end{aligned} \tag{93}$$

Using similar techniques, we can prove that

$$\begin{aligned}
\sum_{t=1}^{T} \sum_{k=1}^{N} \tau(t-1, m_x, \alpha_Q) \|x_t\|_{(\Sigma_{t-1}^Q)^{-1}} O_{t,k}^Q &\leq \tau(T-1, m_x, \alpha_Q) \sqrt{\frac{TN^2 m_x \log\left(1 + \frac{TN}{m_x\alpha_Q}\right)}{\alpha_Q \log(1 + \alpha_Q^{-1})}}, \\
&\leq \chi(T, m_x, \alpha_Q), \\
\sum_{t=1}^{T} \sum_{k=1}^{N} \tau(t-1, m_x^2, \alpha_W) \|z_t\|_{(\Sigma_{t-1}^Q)^{-1}} O_{t,k}^W &\leq \tau(T-1, m_x^2, \alpha_W) \sqrt{\frac{TN^2 m_x^2 \log\left(1 + \frac{TN}{m_x^2\alpha_Q}\right)}{\alpha_W \log(1 + \alpha_W^{-1})}} \\
&\leq \chi(T, m_x^2, \alpha_W).
\end{aligned} \tag{94}$$

On the other hand,

$$\sum_{t=1}^{T} \mathbb{E}\left[\left(\text{Revenue}\left(\boldsymbol{\sigma}_t^*; \boldsymbol{\lambda}_t, q_t, s_t\right) - \text{Revenue}\left(\boldsymbol{\sigma}_t; \boldsymbol{\lambda}_t, q_t, s_t\right)\right) \mathbb{I}\left[\bar{\xi}_{t-1}\right]\right]$$

$$\leq \text{Revenue}\left(\boldsymbol{\sigma}_t^*; \boldsymbol{\lambda}_t, q_t, s_t\right) \sum_{t=1}^{T} \mathbb{E}\left[\mathbb{I}\left[\bar{\xi}_{t-1}\right]\right] \tag{95}$$

$$\leq \frac{r_{\max}}{\epsilon_Q} \sum_{t=1}^{T} \frac{3}{t^2} \leq \frac{C_1 r_{\max}}{\epsilon_Q},$$

for a constant $C_1$. Now the claim follows after combining Equations (82), (93), (94), and (95). $\qquad \square$