# OpenReview forum: "Product Ranking for Revenue Maximization with Multiple Purchases"
_NeurIPS.cc/2022/Conference — NeurIPS 2022 Accept_

### Official Review · Reviewer_tuu8 · 2022-06-29

**Rating:** 4
**Confidence:** 4
**Soundness:** 2 fair
**Presentation:** 2 fair
**Contribution:** 2 fair

**Summary:**

The paper develops vanilla and contextual bandit approaches to maximize revenue given an unknown parametric customer that purchases multiple products.


**Questions:**

See above.

**Limitations:**

Yes.

**Strengths And Weaknesses:**

One of the major issues with this paper is the extremely light and poor comparison with prior work that diminishes the key contributions. Firstly, multiple purchases have already been modeled in the OR literature that are similar to the MNL (e.g., see "Multi-Purchase Behavior: Modeling and Optimization" by Tulabandhula et al, "Assortment Optimization Under Multiple-Discrete Customer Choices" by Zhang et al, etc). Of course, some of these works focus on the revenue maximization problem in the offline setting. Given this context, the multiple purchase contribution looks like yet another model among this growing collection.

For the online setting, in the non-contextual case,there have been many works that follow the same template (define how the user parametrically interacts with a sequence of lists) such as "Thompson sampling for combinatorial semi-bandits" by Wang et al. and "Dynamic learning of sequential choice bandit problem under marketing fatigue" by Cao et al. Especially the latter work shows almost the same model (sans the budget constraint and multi purchase) with near identical results (characterization of the optimal solution in the offline known behavior case) and deriving regret bounds (albeit sparingly for the contextual case in this particular prior work). Given that this type of proof strategy is well known at this point, it is important to highlight what makes the construction of the algorithm and its analysis difficult.

These two issues make one believe that the contributions are incremental and the authors should make an effort to highlight differences.

---

> ### Author Response · Authors · 2022-08-02
> **Response to Reviewer tuu8**
>
> Thank you for your reviewing efforts and constructive comments. We address the concerns point by point.
>
> > *Comment 1: Comparison with previous works on multiple purchases*
>
> Thanks for indicating these papers and we will discuss them thoroughly in the final version of our paper.
>
> The mentioned papers [1, 2] both considered the multiple purchases setting based on MNL. As a result, **they did not take the effect of ranking into consideration**. Specifically, they characterized the probability of purchasing a set of products according to products' and customers' features only. However, in this work, we consider the setting where customers view the product list sequentially. This is a realistic setting since customers can not obtain the full product list on both websites and smartphones and they always view the products from the top to the bottom of the list. Many works are also based on this setting [3, 4]. As a result, the way to rank products has great effects on the purchasing behaviors of customers, making our work differ fundamentally from theirs.
>
>
> > *Comment 2: Comparison with previous works on the online setting*
>
> - **Compared with other UCB-like online algorithms (e.g., [3, 4])**, the main differences lie in the extra parameter $s\_t$ (to characterize the distribution of the purchase budget) we introduce in the customer choice model and hence the algorithms and techniques on the regret analysis.
>     - Firstly, our online algorithms are all based on the optimal ranking policy if the customers' characteristics are known (demonstrated in Section 4). As a result, the optimal ranking policy has fundamental differences from previous works [3, 4, 5] due to the newly introduced parameter $s\_t$.
>     - Secondly, all UCB-like algorithms share a similar protocol (i.e., optimism in the face of uncertainty principle) while the main differences lie in how to estimate the parameters (i.e., the customer's characteristics in our setting) and how to choose the optimal policy in the uncertainty set of the parameters. To be specific, firstly, we estimate the parameters based on Lemma 5.1 and provide the uncertainty set of the parameters in Propositions 5.2 and 5.4. Secondly, the optimal ranking policy in the uncertainty set is chosen based on Propositions B.1 and B.2 in the appendix of our paper. These two steps become more difficult since we introduce the extra parameter $s\_t$. By contrast, existing works do not take the purchase budget into account and can not be adopted directly.
> - **There are also other online algorithms based on Thompson sampling (e.g., [6, 7])**. Both types of algorithms (including algorithms based on UCB or Thompson sampling) can effectively balance the trade-off between exploration and exploitation in online settings. While we study a novel online setting, no existing online algorithms could be adopted here directly. As a result, we focus on UCB-like algorithms in this paper and leave the extension to Thompson sampling algorithms as future work.
> - **Compared with other works on multiple purchases (e.g., [1, 2])**, their algorithms can not be adopted directly here. Firstly, their customer choice models do not take the effect of ranking into consideration, which makes the optimal ranking policy fundamentally different. Secondly, these works focused on designing proper offline algorithms and did not consider the online settings.
>
> [1] Tulabandhula et al. "Multi-purchase behavior: Modeling and optimization." arXiv preprint. 2020.
>
> [2] Zhang et al. "Assortment Optimization Under Multiple-Discrete Customer Choices." Available at SSRN. 2021.
>
> [3] Cao et al. "Dynamic learning of sequential choice bandit problem under marketing fatigue." AAAI. 2019.
>
> [4] Chen et al. "Revenue maximization and learning in products ranking." EC. 2021.
>
> [5] Liang et al. "Sequential dynamic event recommendation in event-based social networks: An upper confidence bound approach." Information Sciences. 2021.
>
> [6] Wang et al. "Thompson sampling for combinatorial semi-bandits." ICML. 2018.
>
> [7] Ferreira et al. "Online network revenue management using thompson sampling." Operations research. 2018.

---

### Official Review · Reviewer_UNt8 · 2022-07-11

**Rating:** 6
**Confidence:** 3
**Soundness:** 4 excellent
**Presentation:** 4 excellent
**Contribution:** 3 good

**Summary:**

The paper studies the problem of product ranking in the following setting (1) customers may purchase more than one product, and (2) customers quit after viewing/purchasing a certain number of products. They first characterize the optimal ranking policy when the customer parameters are known. The paper then investigates the case when customer parameters are unknown. Two UCB-like algorithms with O(\sqrt{T}) regrets are proposed, one for the non-contextual setting (i.e., all customers share the same parameters), and the other for the contextual setting (i.e., customers have personalized parameters). Finally, the effectiveness of the algorithms is tested by simulations.


**Questions:**

1. Could you briefly explain the technical challenges and contributions compared to Liang et al. (2021)?

**Limitations:**

Yes.

**Strengths And Weaknesses:**

Strength:
The paper identifies some limitations of the previous literature and considers a setting that is an “intersection” of several previous setups (mainly Liang et al. (2021), Chen et al. (2021), and Cao and Sun (2019)), which makes their setting more general or realistic compared to each of the previous papers. The paper is complete in the sense that it investigates all the scenarios that have been previously considered (known parameter/non-contextual online setting/contextual online setting) and provides both theoretical analysis and experimental results. I did not check the whole proof, but the theoretical results seem solid and it seems non-trivial to derive the results in this new setting.

Weakness:
For the conceptual contribution, this is an incremental work based on several previous papers. The main concepts and ideas have appeared in previous literature. For the experimental result in the main text, the baselines are some previous methods that are designed under different assumptions. I am not sure how meaningful it is to compare these methods when the synthetic customer data are generated according to a new model.

Disclaimer:
My evaluation of the theoretical contribution may not be accurate because I was not able to access the full pdf of Liang et al. (2021), which seems to contain quite similar results according to the outline.

---

> ### Author Response · Authors · 2022-08-02
> **Response to Reviewer UNt8**
>
> Thank you for your reviewing efforts and constructive comments. We address the concerns point by point.
>
> > *Comment 1: The paper identifies some limitations of the previous literature and considers a setting that is an “intersection” of several previous setups, which makes their setting more general or realistic compared to each of the previous papers. Yet, for the conceptual contribution, this is an incremental work based on several previous papers.*
>
> As mentioned in Line 23-30, most related works in the context of ranking products consider the single purchase setting. Several other works consider the multiple purchases setting while their targets are not revenue maximization [1, 2] or they assume that customers always continue viewing the product list after purchasing any product [3]. By contrast, we consider the purchase budget and generalize the existing customer choice models to a more realistic setting, which is novel.
>
> > *Comment 2: The baselines are some previous methods that are designed under different assumptions.*
>
> - Firstly, because we consider a novel setting and no existing works could fit perfectly here, we choose the closest works [3, 4] as baselines to demonstrate the effectiveness of our algorithms.
> - Secondly, our setting is more realistic and general and their works [3, 4] consider the special cases of our setting. We implement their methods based on their assumptions to ensure the solidness of their results so that the comparison between our method and theirs is reliable and rational.
>
> > *Comment 3: Could you briefly explain the technical challenges and contributions compared to Liang et al. (2021)?*
>
> The main differences between our work and Liang et al. [3] lie in the problem setting (especially the customer choice model) and hence the algorithms and techniques to achieve the regret analysis.
>
> - **The differences in settings** have the following two aspects.
>     1. The most fundamental difference between our setting and theirs is **the purchase budget that constrains the total number of purchases of each customer**. As demonstrated in Line 28-30, [3] assumed that customers always continue viewing the product list after purchasing any product, which is not practical in real scenarios. By contrast, the purchase budget proposed in our paper leads to a different and more practical customer choice model. In addition, our model introduces an extra parameter (i.e., the parameter $s\_t$ that characterizes the distribution of the purchase budget), which makes our problem more challenging.
>     2. [3] assumed the online retailer will stop recommending products with a certain probability $1 - w\_t$ to avoid the cost brought by marketing fatigue (i.e., overexposure to unwanted marketing messages [4]). As a result, the parameter $w\_t$ is a part of the ranking policy instead of customers' characteristics and should be optimized to achieve maximal revenue. In our paper, we focus on the multiple purchases with budgets setting and leave the extension to the case with the marketing fatigue as future work.
> - **The differences in techniques** have the following two aspects.
>     1. For the optimal ranking policy given customers’ characteristics, our optimal ranking policy is based on both the newly introduced parameter $s\_t$ and other customers' characteristics (i.e., the attention span and the purchase probabilities) as shown in Theorem 4.1. By contrast, they did not take the purchase budget into account so their optimal policy should be based on other customers' characteristics only, which differs from the result in Theorem 4.1 of our paper.
>     2. For the online parts, all UCB-like algorithms share a similar protocol (i.e., optimism in the face of uncertainty principle) while the main differences lie in how to estimate the parameters (i.e., the customer's characteristics in our setting) and how to choose the optimal policy in the uncertainty set of the parameters. Specifically, we first estimate the parameters (including $s\_t$) based on Lemma 5.1 and provide the uncertainty set of the parameters in Propositions 5.2 and 5.4. Then the optimal ranking policy in the uncertainty set is chosen based on Propositions B.1 and B.2 in the appendix of our paper. By contrast, the parameter $w\_t$ introduced in [3] is a part of the ranking policy instead of customers' characteristics and need not be estimated from historical data. As a result, our online algorithms are more challenging due to the existence of the extra parameter $s\_t$ in the customer choice model.
>
> [1] Cao et al. "Fatigue-aware bandits for dependent click models." AAAI. 2020.
>
> [2] Ferreira et al. "Learning to rank an assortment of products." Management Science. 2022.
>
> [3] Liang et al. "Sequential dynamic event recommendation in event-based social networks: An upper confidence bound approach." Information Sciences. 2021.
>
> [4] Cao et al. "Dynamic learning of sequential choice bandit problem under marketing fatigue." AAAI. 2019.

---

> > ### Comment · Reviewer_UNt8 · 2022-08-09
> > **Thanks for the reponses**
> >
> > Thank you for the explanations.

---

> > > ### Author Response · Authors · 2022-08-09
> > > **Thanks for the response**
> > >
> > > We sincerely appreciate your response. We will incorporate your feedback and discuss the relationship with Liang et al. in the final version of the paper. Thank you for your questions and suggestions again!

---

### Official Review · Reviewer_PDX6 · 2022-07-11

**Rating:** 7
**Confidence:** 3
**Soundness:** 3 good
**Presentation:** 3 good
**Contribution:** 4 excellent

**Summary:**

The authors study the problem of product ranking based on a more realistic situation where customers can have multiple purchases and continues viewing products even after buying a product, both being subject to an overall attention span and overall budget. They characterize the optimal ranking policy and design UCB-like algorithms with a provable \tildeO(\sqrt{T}) regret based on both contextual and non-contextual situations. They also show extensive experiments to prove their efficacy on the simulated and real-world datasets.


**Questions:**

See above.

**Strengths And Weaknesses:**

The paper is very well written (though with minor typos (see below)) and is quite easy to follow. The related work section throws good light into the existing literature on which this work is based. The thorough theoretical section is impressive and the results look good.

The major comment is on the time complexity of the algorithms in practice. Section 4 shows the time complexity as O(N Log N). Please add a discussion on how Algorithms 1 and 2 can put into practice with similar time complexity and appropriate parallelization for each t.

Minor typos:
1. Line 53, 314: rankling -> ranking
2. Line 210: management

---

> ### Author Response · Authors · 2022-08-02
> **Response to Reviewer PDX6**
>
> Thank you for your reviewing efforts and constructive comments. We address the concerns point by point.
>
> > *Comment 1: The major comment is on the time complexity of the algorithms in practice. Please add a discussion on how Algorithms 1 and 2 can put into practice with similar time complexity and appropriate parallelization for each t.*
>
> We discuss the time complexity of the algorithms as follows.
>
> - For Algorithm 1 in the non-contextual setting, the statistics in Line 6 of Algorithm 1 can be updated at $O(N)$ time complexity. Combining the $O(N \log N)$ complexity to calculate the optimal ranking policy according to Theorem 4.1, the overall time complexity for each $t$ is $O(N \log N)$. As a result, the algorithm in this setting does not need parallelization.
> - For Algorithm 2 in the contextual setting, the major overhead is on the calculation of the matrix inversion in Lines 5 and 9 of Algorithm 2. The corresponding time complexity is $O(m\_x^6 + m\_y^2)$ where $m\_x$ is the dimension of customers' features and $m\_y$ is the dimension of the joint feature of customers and products. As a result, the overall time complexity for each $t$ is $O(N \log N + m\_x^6 + m\_y^2)$. However, the inverse of matrices can be calculated effectively under parallelization [1]. As a result, the actual time complexity can be much smaller.
>
> > *Comment 2: the minor typos.*
>
> Thanks for indicating these mistakes, and we will correct them in the final version of the paper.
>
> [1] Quintana et al. "A note on parallel matrix inversion." SIAM Journal on Scientific Computing. 2001.

---

### Official Review · Reviewer_oMCA · 2022-07-12

**Rating:** 5
**Confidence:** 4
**Soundness:** 3 good
**Presentation:** 3 good
**Contribution:** 2 fair

**Summary:**

This paper studies a multi-product purchase problem. Authors design an optimal ranking policy when the online retailer can precisely model customers’ behaviors. Most of the existing work assume that each customer purchases at most one product and stops browsing immediately. This paper proposes a more realistic setting that customers view the product list sequentially and purchase a product when its utility exceeds a certain threshold with a certain purchase budget. In both non-contextual and contextual settings, they provide UCB-like algorithms and theoretically prove that the algorithms achieve \tilde{O}(\sqrt{T}) regrets in both settings.

**Questions:**

1.	According to the purchase probability specified in Eq. (1), the purchase probabilities towards different products are independent of each other. How do you verify this assumption? Please also discuss the corresponding applications.

2.	Could you emphasize the major technical contribution of the online algorithm? For example, how does the analysis differ from the existing work? Which part of the single purchase problem cannot be used in this problem? If it can be emphasized before the main results, it will help to improve the paper.

3.	It is assumed that $V_t$ follows the geometric distribution with parameter 1-q_t and B_t follows the geometric distribution with parameter 1-s_t. I think Theorem 4.1 heavily relies on this assumption. Can the result be generalized to other distributions?

4.	In Section 5, two Theorems specify the regret upper bounds for both non-contextual and contextual problems. Could you also prove that they are both tight?


**Limitations:**

The attention span and the purchase budget are both limited to the geometric distribution. It would improve the paper if they can be generalized to other distributions.

**Strengths And Weaknesses:**

Strengths:
This paper considers a more practical setting where a single customer can make multiple purchases with certain purchase budget and attention span.
Theorem 4.1 provides a nice structure of the optimal ranking policy.
The study of this ranking problem is comprehensive. It studies both the non-contextual and contextual version.
The paper is nicely written.

Weaknesses:

Although the paper considers a more practical problem, it still has some limitations: the paper assumes that V_t follows the geometric distribution with parameter 1-q_t and B_t follows the geometric distribution with parameter 1-s_t. Each item costs one unit of budget.

---

> ### Author Response · Authors · 2022-08-02
> **Response to Reviewer oMCA (Part 2 / 2)**
>
> > *Comment 3: It is assumed that $V\_t$ follows the geometric distribution with parameter $1-q\_t$ and $B\_t$ follows the geometric distribution with parameter $1-s_t$. Each item costs one unit of budget. I think Theorem 4.1 heavily relies on this assumption. Can the result be generalized to other distributions?*
>
> - Theorem 4.1 is practical because the assumptions are mild and they can cover common realistic scenarios.
>     - Firstly, **the assumption that the attention span obeys the geometric distribution is common in related literature**. The cascading models [1] are all based on the assumption that each item in a list is less likely to be viewed by the user with an exponential decay [2], making the attention span obey the geometric distribution. While cascading models are widely adopted (e.g., [6, 7]) to characterize customers' behaviors, it is rational and general to assume the attention span obeys the geometric distribution.
>     - Secondly, **the numbers of purchases/clicks are fitted well with the geometric distribution in reality**. [3] showed that the data obtained from Overture during 2003 (including data from Yahoo!, MSN, and AltaVista) are fitted well ($R^2=0.997$) by an exponential decay model (equivalently a geometric distribution). As a result, it is rational and general to assume the purchase budget follows the geometric distribution.
> - We further discuss the possible relaxations on the assumptions as follows.
>     - **Relaxing the unit cost assumption**. Suppose that each product has a $e_n$ cost and the customer will stop viewing if the total cost exceeds the purchase budget. By assuming the geometric distribution conditions, following similar proof techniques, the optimal ranking policy in Theorem 4.1 becomes
>     $$
>     \frac{\lambda\_{\sigma\_i^\*}r\_{\sigma\_i^\*}}{1 - q + q\left(1-s^{e\_{\sigma\_i^\*}}\right)\lambda\_{\sigma\_i^\*}} \ge \frac{\lambda\_{\sigma\_{i+1}^\*}r\_{\sigma\_{i+1}^\*}}{1 - q + q\left(1-s^{e\_{\sigma\_{i+1}^\*}}\right)\lambda\_{\sigma\_{i+1}^\*}}.
>     $$
>     - **Relaxing the geometric assumption**. The problem becomes NP-hard under this circumstance because [4] studied the reduced version of the problem (customers purchase at most one product and the distribution of attention span can be other distributions.), which is NP-hard. However, proper approximation algorithms may exist (e.g., the algorithms in [4]) and we leave them as future work.
>
> > *Comment 4: In Section 5, two Theorems specify the regret upper bounds for both non-contextual and contextual problems. Could you also prove that they are both tight?*
>
> **The regret bounds match the best-achievable rate in the literature** as we demonstrated in the corresponding remarks of the theorems. In particular, we study a general case of the previous work [7] and achieve regret bounds with the same rate, indicating the tightness and reliability of our result. In detail, the most important parameters include the number of products $N$ and the number of customers $T$. We highlight the main points as follows.
>
> - For the non-contextual setting, firstly, the regret grows at $\tilde{O}(\sqrt{T})$, which is a standard result in online learning algorithms [5]. Secondly, the regret bound is linearly related to $N$, which is the same with the results in cascade bandit models [6] and revenue management literature [4, 7]. As a result, the overall regret bound is $\tilde{O}(N\sqrt{T})$, which matches the regret bound of the reduced setting considered in [7].
> - For the contextual setting, firstly, the regret grows at $\tilde{O}(\sqrt{T})$, a standard result in online learning algorithms [5]. Secondly, the regret depends on $\tilde{O}(N)$. This shares the same result with [4], which considered similar linear assumptions on the customers' characteristics.
>
> [1] Craswell et al. "An experimental comparison of click position-bias models." WSDM. 2008.
>
> [2] Breese et al. "Empirical analysis of predictive algorithms for collaborative filtering." UAI. 1998.
>
> [3] Feng et al. "Implementing sponsored search in web search engines: Computational evaluation of alternative mechanisms." INFORMS Journal on Computing. 2007.
>
> [4] Chen et al. "Revenue maximization and learning in products ranking." EC. 2021.
>
> [5] Slivkins, Aleksandrs. "Introduction to multi-armed bandits." Foundations and Trends in Machine Learning. 2019.
>
> [6] Zong et al. "Cascading bandits for large-scale recommendation problems." UAI. 2016.
>
> [7] Cao et al. "Dynamic learning of sequential choice bandit problem under marketing fatigue." AAAI. 2019.

---

> ### Author Response · Authors · 2022-08-02
> **Response to Reviewer oMCA (Part 1 / 2)**
>
> Thank you for your reviewing efforts and constructive comments. We address the concerns point by point.
>
> > *Comment 1: According to the purchase probability specified in Eq. (1), the purchase probabilities towards different products are independent of each other. How do you verify this assumption? Please also discuss the corresponding applications.*
>
> - The assumption in Equation (1) is that **the conditional purchase probabilities are independent of each other**. That means if a customer $t$ views the product $k$, the purchase probability $\lambda\_{t,k}$ is independent of other products and the position of the product in the ranking list. **This assumption is common and standard in related literature [1-6].** On the one hand, several customer choice models are characterized based on this assumption, including the cascading model [1] and the click chain model [2]. On the other hand, various works have been proposed based on these choice models, as well as the underlying independent assumption, to maximize the revenue [3, 4] or the number of clicks [5, 6].
> - We highlight that **the actual purchase probability of the product $k$ in the ranking list depends both on other products and the position of the product $k$**. Because of the existence of the attention span and purchase budget, the probability that a customer views the product $k$ tends to decrease when the product appears later in the ranking list. In addition, if the products appeared before the product $k$ have high (low) purchase probabilities, it will be easier (harder) for the customer to meet the purchase budget and stop viewing, making the product $k$ less (more) likely to be viewed. As a result, the purchase probability of the product $k$ will be affected.
> - **The assumption in Equation (1) can be satisfied in real-world scenarios**. For example, in online retailer platforms or web click scenarios, as long as the customer views a product/web, the purchase/click probability only depends on his or her own interest in the product/web, making the conditional purchase/click probability independent of other variables. These scenarios are well studied in related literature (e.g., [1,3]).
>
> > *Comment 2: Could you emphasize the major technical contribution of the online algorithm? For example, how does the analysis differ from the existing work? Which part of the single purchase problem cannot be used in this problem?*
>
> Generally speaking, we consider the more realistic setting (i.e., the multiple purchases with budgets setting) and introduce an extra parameter $s\_t$ to characterize the distribution of the purchase budget in the customer choice model, which makes the algorithms more challenging. Details are provided as follows.
>
> - Firstly, our online algorithms are all based on the optimal ranking policy if the customers' characteristics are known (demonstrated in Section 4). **Because of the extra parameter $s\_t$, the optimal ranking policy has fundamental differences from previous works on single purchase settings [3, 4]**.
> - Secondly, all UCB-like algorithms share a similar protocol (i.e., optimism in the face of uncertainty principle) while the main differences lie in how to estimate the parameters (i.e., the customer's characteristics in our setting) and how to choose the optimal policy in the uncertainty set of the parameters. To be specific, firstly, we estimate the parameters based on Lemma 5.1 and provide the uncertainty set of the parameters in Propositions 5.2 and 5.4. Secondly, the optimal ranking policy in the uncertainty set is chosen based on Propositions B.1 and B.2 in the appendix of our paper. **These two steps are more challenging compared with previous methods due to the extra parameter $s\_t$**. By contrast, existing works on single purchase settings can not be adopted directly here.
>
> [1] Craswell et al. "An experimental comparison of click position-bias models." WSDM. 2008.
>
> [2] Guo et al. "Click chain model in web search." WWW. 2009.
>
> [3] Chen et al. "Revenue maximization and learning in products ranking." EC. 2021.
>
> [4] Cao et al. "Dynamic learning of sequential choice bandit problem under marketing fatigue." AAAI. 2019.
>
> [5] Kveton et al. "Cascading bandits: Learning to rank in the cascade model." ICML. 2015.
>
> [6] Katariya et al. "DCM bandits: Learning to rank with multiple clicks." ICML. 2016.

---

### Author Response · Authors · 2022-08-08
**Thank all reviewers for the insightful suggestions**

Dear reviewers,

We appreciate your invaluable feedback and constructive suggestions. We are wondering whether our response addressed your concerns. If you have any additional comments, please let us know. We would be happy to address them.

---

### Meta-Review · Area_Chair_t1U9 · 2022-08-28

**Recommendation:** Accept
**Confidence:** Less certain

**Metareview:**

The paper studies the problem of choosing a ranked list of products to show to consumers in a regret minimization model. Consumers are assumed to follow a certain search rule to purchase a subset of presented products, and the goal is to maximize the revenue of the product listing under this search model. The model makes certain assumptions of the previously studied models somewhat more realistic, for example, it allows the consumer to purchase more than one product. The main result is a UCB-like algorithm with the regret of O(sqrt(T)).

On the negative side: Even though the model claims to make the model more realistic than the other models previously studied in the literature, I still find the model quite stylized, and would not call it a practical model for capturing consumer behavior. For example, in the real world, one would expect the consumers to compare their options and take this comparison into account when selecting which item to purchase, whereas in this paper, the consumer makes a probabilistic decision on each item it sees independent of the other items (only conditioning on not having already purchased enough items).

On the positive side, the paper solves a meaningful and non-trivial, though somewhat stylized problem, and the results are interesting, at least from a theoretical point of view.

For these reasons, I'm leaning to accept this paper, if it fares well in comparison with other papers on the borderline.

**Award:**

No

---

### Decision · Program_Chairs · 2022-09-14

Accept